# CRIBO 🐘: SELF-SUPERVISED LEARNING VIA CROSS-IMAGE OBJECT-LEVEL BOOTSTRAPPING

**Tim Lebailly**[1]*    **Thomas Stegmüller**[2]*    **Behzad Bozorgtabar**[2,3]
**Jean-Philippe Thiran**[2,3]    **Tinne Tuytelaars**[1]
[1]KU Leuven    [2]EPFL    [3]CHUV
[1]{firstname}.{lastname}@esat.kuleuven.be    [2]{firstname}.{lastname}@epfl.ch

## ABSTRACT

Leveraging nearest neighbor retrieval for self-supervised representation learning has proven beneficial with object-centric images. However, this approach faces limitations when applied to scene-centric datasets, where multiple objects within an image are only implicitly captured in the global representation. Such global bootstrapping can lead to undesirable entanglement of object representations. Furthermore, even object-centric datasets stand to benefit from a finer-grained bootstrapping approach. In response to these challenges, we introduce a novel **Cr**oss-**I**mage Object-Level **Bo**otstrapping method tailored to enhance dense visual representation learning. By employing object-level nearest neighbor bootstrapping throughout the training, CrIBo emerges as a notably strong and adequate candidate for in-context learning, leveraging nearest neighbor retrieval at test time. CrIBo shows state-of-the-art performance on the latter task while being highly competitive in more standard downstream segmentation tasks. Our code and pretrained models are publicly available at https://github.com/tileb1/CrIBo.

## 1 INTRODUCTION

Over the past few years, the field of artificial intelligence has experienced significant advancements, primarily driven by the democratization of deep learning techniques (Krizhevsky et al., 2012). Self-supervised learning (SSL) stands out as one of the major factors contributing to the success of deep learning. Indeed, SSL has opened the door to the training of large foundation models pretrained on massive amounts of uncurated data. In natural language processing (NLP), this large-scale task-agnostic pretraining has been the key to unlocking general-purpose features that perform well on various downstream tasks. Conversely, in computer vision, task-specific models and finetuning remain the *de facto* choice, in part due to the lack of consensus around the pretraining pretext task, such as is the case in NLP with masked language modeling (MLM).

Nonetheless, Balaževič et al. (2023) showed that under the auspices of their contextual pretraining, dense nearest neighbor retrieval performed comparably with task-specific models while being

---

*denotes equal contribution.

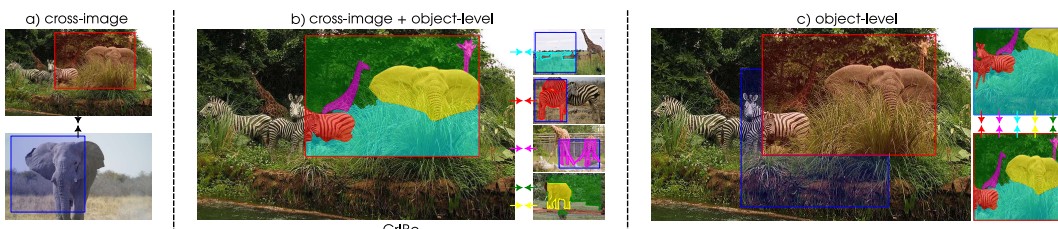

Figure 1: **Positioning of CriBO in the landscape of self-supervised learning. a)** Illustration of the cross-image self-supervision concept. **c)** Depiction of object-level self-supervision. **b)** CrIBo benefits from both learning paradigms.

significantly more efficient. The contextualization mechanism ensures the alignment of the pretraining phase with dense nearest neighbor evaluation by leveraging within and across image attention. Nevertheless, this mechanism only partially fulfills the alignment objective as *i)* the cross-image attention mechanism occurs from **local to global** features and *ii)* consistency is only encouraged between contextualized **image-level** representations of views from the **same image**. To mitigate these limitations, we propose to **explicitly** enforce cross-image consistency between object-level representations as shown in Figure 1.

Most recent self-supervised learning algorithms, *e.g.*, Chen et al. (2020); He et al. (2020); Grill et al. (2020a); Caron et al. (2021), revolve around the objective of learning image-level representations invariant to semantic-preserving data augmentations. CrIBo departs from this paradigm in two ways. First, we encourage consistency at the granularity of the object in the images, and second, this consistency is enforced between objects from different images. In doing so, our pretraining aligns well with the objective of obtaining general-purpose representations tailored for dense nearest neighbor retrieval. Furthermore, by operating at the object level, CrIBo elegantly mitigates the pitfall of contextual bias Singh et al. (2020), and is therefore compatible with *scene-centric* images which constitute the bulk of web-scale datasets.

Our contributions are as follows. **1)** To the best of our knowledge, CrIBo is the first end-to-end online SSL approach that explicitly enforces cross-image consistency at the object/object-part level. **2)** The resulting representations excel in *in-context scene understanding* tasks (Balažević et al., 2023) without compromising the performance on standard segmentation benchmarks. **3)** Moreover, CrIBo is compatible with scene-centric datasets, addressing a gap in the existing cross-image SSL literature.

## 2  RELATED WORKS

**Image-level self-supervision.** Seminal work SimCLR (Chen et al., 2020) has played a pivotal role in promoting the widespread adoption of *cross-view consistency* as a fundamental principle for visual representation learning. However, such pretext task admits trivial solutions and contemporary works have yet to agree on the ideal way to prevent them. One effective and intuitive approach to address this issue involves the use of negative samples (Chen et al., 2020; Hjelm et al., 2018), which effectively mitigates degeneracy but requires the use of large batchsizes. This can be alleviated with a memory bank (He et al., 2020). Self-distillation methods, on the other hand, do not use explicit negatives. They avoid trivial solutions by using asymmetry in the form of *e.g.*, an additional predictor (Chen & He, 2020; Grill et al., 2020b) on one branch, using stop-gradients (Chen & He, 2020; Caron et al., 2021; Li et al., 2021; Grill et al., 2020b), a momentum encoder (Caron et al., 2021; Grill et al., 2020b), *etc*. On the other end of the spectrum, clustering-based methods avoid trivial solutions by regularizing the assignment of the samples across a set of clusters (Asano et al., 2019; Caron et al., 2018; 2020; 2021; Zhuang et al., 2019).

**Localized self-supervision.** A finer-grained self-supervision can be obtained by leveraging the inherent spatial structure of images and by enforcing cross-view consistency at a localized level. This ensures that the resulting features are better aligned with dense downstream tasks. These methods can be broadly categorized based on the level of granularity at which they enforce similarity. One line of work involves enforcing similarity at the feature level, known as the local-level approach, where individual features (or local-representations) are directly contrasted (Wang et al., 2021; Liu et al., 2020; O Pinheiro et al., 2020; Xie et al., 2021b; Lebailly & Tuytelaars, 2022; Bardes et al., 2022). Another approach operates at a coarser level, promoting similarity between semantically coherent groups of features, referred to as the object-level approach (Cho et al., 2021; Hénaff et al., 2021; 2022; Xie et al., 2021a; Seitzer et al., 2022; Stegmüller et al., 2023; Wen et al., 2022).

Alternatively, there have been propositions of dense fine-tuning strategies (Hamilton et al., 2022; Ziegler & Asano, 2022; Yun et al., 2022; Wang et al., 2022; Zadaianchuk et al., 2022). These techniques are designed to enhance models initially trained with an image-level objective (Caron et al., 2021) with localized self-supervision. However, the former cannot be used as stand-alone pretraining methods. Another way of using the inherent spatial structure is by leveraging patch location prediction (Caron et al., 2022) or by asking the model to reconstruct the input image from a corrupted/masked version thereof (He et al., 2022).

**Cross-image self-supervision.** Moving beyond the conventional cross-view consistency inherent in numerous self-supervised methods, image-level bootstrapping harnesses nearest neighbors in the latent space for self-labeling and metric learning (Dwibedi et al., 2021; Koohpayegani et al., 2021; Azabou et al., 2021; Xie et al., 2022). Current bootstrapping methods for representation learning face two challenges. Firstly, the effectiveness of using distance metrics in the latent space as a proxy for semantic closeness is contingent on the quality of the learned representations, especially during the initial stage of pretraining when the encoder is randomly initialized. To address this limitation, adaptive bootstrapping methods have been proposed, aiming to avoid systematic nearest neighbor bootstrapping (Lebailly et al., 2023). Secondly, in the context of scene-centric images, image-level bootstrapping reveals a tendency to entangle object representations, which can have detrimental effects on dense downstream tasks. These issues motivate the introduction of the object-level cross-image self-supervised learning paradigm. To that end, ORL (Xie et al., 2021a) introduced a three-stage procedure to identify pairs of region-of-interests (RoIs) and encourage cross-image consistency between cropped objects at the image-level.

Recently, Balaževic et al. (2023) proposed to leverage attention within and across images. The latter contextualizes the local representations of an image with the global ones from other images to adapt for in-context learning of various scene understanding tasks at test time.

## 3 METHOD

### 3.1 PRELIMINARIES

Before diving into the method, we briefly introduce the terminology used in this paper.

**Dense representation.** A representation that explicitly preserves the spatial dependencies of the input image is referred to as a *dense representation*. We denote such representations by $z \in \mathbb{R}^{H \times W \times d}$ or $\in \mathbb{R}^{HW \times d}$ where $H$ and $W$ are the height and width of the image or a downscaled version thereof and $d$ refers to the dimension of the latent space. The dense representation produced by a convolutional neural network is a stack of output feature maps, whereas a vision transformer yields an ordered sequence of tokens.

**Local representation.** Given a spatial location $n$ out of the $N = HW$ possible ones, the associated slice of the dense representation $z^n \in \mathbb{R}^d$ with $n \in \{1, 2, \cdots, N\}$ is referred to as a *local representation*.

**Object representation.** The *object representation* $c^k \in \mathbb{R}^d$ of the $k^{\text{th}}$ object within an image is derived by aggregating the local representations associated with the corresponding object. We use the term *object* loosely to refer to a semantically coherent region within the image.

**Global representation.** The *global representation*, denoted as $\bar{z} \in \mathbb{R}^d$, is a global aggregation over the spatial dimensions of the dense representation. As an example, this corresponds to the `[CLS]` token in a ViT (Dosovitskiy et al., 2020) or the result of a global average pooling layer in case of a CNN.

**Bootstrapping** Here, the term *bootstrapping*[1] refers to the utilization of the model's current knowledge to enhance its performance further. In the scope of this study, this materializes as leveraging distances in the latent space to find object-level nearest neighbors that can be used to form positive pairs for self-distillation.

### 3.2 OBJECT-LEVEL CROSS-IMAGE BOOTSTRAPPING (CRIBO)

In Section 1, we discuss the need for generalist models capable of performing various downstream tasks without requiring any finetuning. A recent study (Balaževic et al., 2023) demonstrated that this could be achieved by leveraging nearest neighbor retrieval at test time and contextualization

---

[1]The term bootstrapping is employed in the colloquial sense, as opposed to the statistical sense.

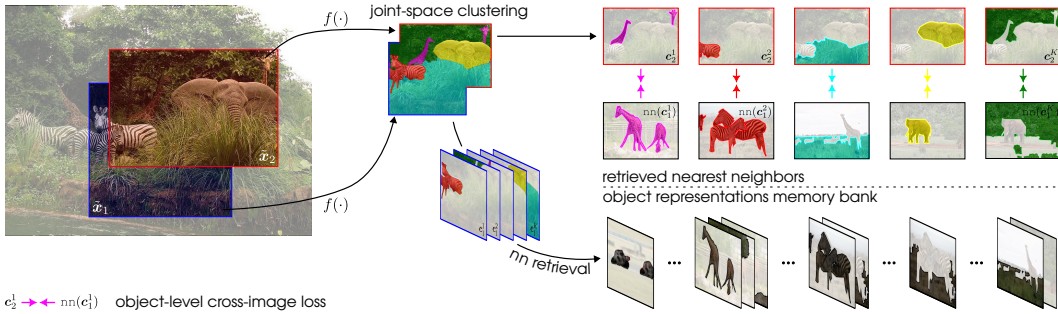

Figure 2: **High-level overview of cross-image object-level bootstrapping (CrIBo).** Given an encoder $f$ and pair of augmented views $\tilde{x}_1$ and $\tilde{x}_2$, object representations $c_i^k$ (depicted as colored object masks) from each view $i$ are computed. Using a memory bank, the nearest neighbors of each object representation $c_1^k$ from the first view are retrieved. A self-supervised consistency loss (depicted as colored arrows) is then enforced between $c_2^k$ and its corresponding retrieved neighbor from the other view $\text{nn}(c_1^k)$.

throughout the training phase. Aligning with this aim, we propose further reducing the gap between the training and test phases by explicitly enforcing consistency between object-level nearest neighbors. The existing SSL studies exploiting nearest neighbors are built solely around global representations and are rooted in the assumption of datasets being object-centric. Hence, a simplistic extension of such frameworks to our proposed scenario is impracticable. To accomplish our goal, the following steps are required:

1. Identify semantically coherent regions within the image to create object-level representations (Sec. 3.2.1).

2. Match pairs of object-level representations across images (Sec. 3.2.2).

3. Apply cross-image consistency between pairs of matching object-level representations (Sec. 3.2.3).

A high-level overview of our proposed method CrIBo can be found in Figure 2.

### 3.2.1 SEMANTICALLY COHERENT IMAGE REGIONS

The first step towards cross-image object-level bootstrapping is to identify semantically coherent image regions within the images. We argue that it is beneficial to use pairs of augmented views for each input image and to have a cross-view correspondence between the objects in both views. Indeed, image-level SSL methods yield significantly worsened performance when training without photometric augmentations (Dwibedi et al., 2021). This is observed when positive pairs originate from the same image and to a lesser extent (but still significant) when they do not. This drop in performance highlights the need to shed away low-level similarities between positive pairs of images with data augmentations. To prevent this from occurring with nearest neighbor (NN) positives, Dwibedi et al. (2021) refrain from enforcing consistency between a retrieved NN and its query. At the image level, this can be easily implemented by leveraging the duplicity of the views. More specifically, given two augmented views $\tilde{x}_1$ and $\tilde{x}_2$ as well as their corresponding global-representation $\bar{z}_1$ and $\bar{z}_2$, a similarity constraint is enforced between $\bar{z}_1$ and $\text{nn}(\bar{z}_2)$ where $\text{nn}(\cdot)$ denotes the nearest neighbor operator. Conversely, at the object level, this requires a cross-view correspondence between object representations from both views. To that end, we rely on an online joint-space clustering algorithm from Stegmüller et al. (2023).

**Joint-space clustering.** The clustering algorithm takes as input the dense representations $z_1 \in \mathbb{R}^{N \times d}$ and $z_2 \in \mathbb{R}^{N \times d}$ associated with the two augmented views and concatenates them along the token axis to obtain the dense representation of the joint-view, $z_{\text{cat}} \in \mathbb{R}^{2N \times d}$. The $2N$ tokens are subsequently partitioned into $K$ clusters to produce semantically coherent and spatially compact clusters. A hyperparameter $\lambda_{\text{pos}}$ modulates the importance of the spatiality against that of the semantic (see Tab. 4). The resulting assignment is encoded in a matrix $\mathbf{Q}^* \in \mathbb{R}^{2N \times K}$, which can

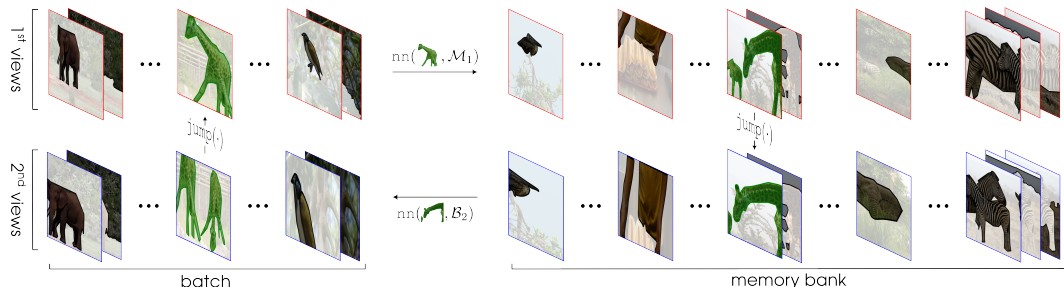

Figure 3: **Illustration of cycle-consistent matchings**. Such matchings are invariant to data augmentations and reciprocal, loosely speaking.

be split in two to obtain the view-wise assignments $\mathbf{Q}_1^*, \mathbf{Q}_2^* \in \mathbb{R}^{N \times K}$. Here $\mathbf{Q}_i^*(n, k)$ denotes a soft-assignment of token $n$ to object-cluster $k$ in view $i$. These can in turn be used to compute the $K$ object representations $\boldsymbol{c}_i \in \mathbb{R}^d$ in each view $i$ as follows:

$$\boldsymbol{c}_i^k = \sum_{n \in [N]} \mathbf{Q}_i^*(n, k) \cdot \boldsymbol{z}_i^n, \qquad i \in \{1, 2\} \tag{1}$$

Note that the columns of $\mathbf{Q}_i^*$ undergo $l_1$ normalization, rendering the object representations as affine combinations of local representations. It can be observed that the cross-view correspondence is provided for free as the object representation $\boldsymbol{c}_1^k$ from the first view is matched to $\boldsymbol{c}_2^k$ in the second one as a consequence of the joint-space clustering. In practice, it can be the case that some objects are exclusive to one of the two views *i.e.*, $\sum_{n \in [N]} \mathbf{Q}_i^*(n, k) = 0$. In that case, the object $k$ is simply discarded from the view where it is present. In Section 3.2.2, we show that such cross-view correspondence can be harnessed to improve the training stability through a bootstrapping criterion.

### 3.2.2 CROSS-IMAGE OBJECT MATCHINGS

In Section 3.2.1, we detail the methodology for identifying objects within the views of a given image and establish the strategy for their alignment from one view to another. Subsequent to this step is the critical phase of aligning such objects across images. For reasons that will be delineated in the next paragraph, we rely on a pair of memory banks $\mathcal{M}_1$ and $\mathcal{M}_2$, which are populated with object representations extracted from the first and second views during previous iterations. As such, the memory banks $\mathcal{M}_i$ can be queried with an object representation $\boldsymbol{c}_i^k$ to retrieve its nearest neighbor:

$$\operatorname{nn}(\boldsymbol{c}_i^k, \mathcal{M}_i) \triangleq \underset{\boldsymbol{c} \in \mathcal{M}_i}{\arg\max} \frac{\boldsymbol{c}^\top \boldsymbol{c}_i^k}{\|\boldsymbol{c}\|_2 \|\boldsymbol{c}_i^k\|_2} \tag{2}$$

Here, we consider $\operatorname{nn}(\boldsymbol{c}_1^k, \mathcal{M}_1)$ and $\boldsymbol{c}_1^k$ as cross-image object-level matches. A visualization of cross-image object-level matches can be found in Appendix C.5 (Fig. 4 and Fig. 5).

**Cycle-consistent matchings.** At the beginning of the pretraining phase, when the encoder is randomly initialized, relying on Equation (2) to identify cross-image object-level matches is not well grounded. Indeed, this straightforward approach assumes that the representations of similar objects are close in the embedding space, an assumption that does not hold at the initialization. Therefore, enforcing consistency between such pairs of semantically distinct objects should be avoided. To that end, we design a *semantic closeness* condition, which is only fulfilled when the matches are established based on high-level features, thus enabling the discarding of invalid positive pairs that do not share a semantic similarity.

At a high-level, we consider a pair of objects as a valid positive pair if they are the nearest neighbors of each other. In practice, the criterion is implemented as follows. Starting from an object representation $\boldsymbol{c}_1^k$ in the batch of first views, its nearest neighbor $\operatorname{nn}(\boldsymbol{c}_1^k, \mathcal{M}_1)$ in the queue is found using Equation (2). To ensure that the criterion is invariant to data augmentation, we then switch to the queue of second views via the $\operatorname{jump}(\cdot)$ operator:

$$\operatorname{jump}(\boldsymbol{c}_1^k) \triangleq \boldsymbol{c}_2^k, \qquad \operatorname{jump}(\boldsymbol{c}_2^k) \triangleq \boldsymbol{c}_1^k \tag{3}$$

From this step, we obtain $\mathtt{jump}(\mathtt{nn}(\boldsymbol{c}_1^k, \mathcal{M}_1))$ and retrieve its NN in the batch of second views $\mathtt{nn}(\mathtt{jump}(\mathtt{nn}(\boldsymbol{c}_1^k, \mathcal{M}_1)), \mathcal{B}_2)$. The final step consists of switching back to the batch of first views and verifying if the resulting object is the same as the one we started from. More formally:

$$\mathcal{O}(\boldsymbol{c}_i^k) \triangleq \begin{cases} true & \text{if} \quad \boldsymbol{c}_i^k = \mathtt{jump}(\mathtt{nn}(\mathtt{jump}(\mathtt{nn}(\boldsymbol{c}_i^k, \mathcal{M}_i)), \mathcal{B}_j)) \\ false & \text{otherwise} \end{cases} \qquad i, j \in \{1, 2\}, i \neq j \qquad (4)$$

A depiction of cycle-consistent matchings is provided in Figure 3.

### 3.2.3 SELF-SUPERVISED TRAINING OBJECTIVES

At this point, we have all the necessary components to apply three distinct types of self-supervision: cross-view object-level, cross-image object-level, and global cross-image. All levels of self-supervision in our method CrIBo are based on the self-distillation mechanism proposed by Caron et al. (2021), which employs a student-teacher pair of siamese networks $f_s$ and $f_t$, respectively. Given two augmented views $\tilde{\boldsymbol{x}}_1$ and $\tilde{\boldsymbol{x}}_2$, global representations and object representations can be obtained for each view and each network within the siamese pair.

**Cross-View Object-Level Self-Supervision.** The object representations from the teacher and student are fed to their respective head $h_s$ and $h_t$ to output probability mass functions whose sharpness can be modulated via temperature parameters ($\tau_s$ and $\tau_t$). For the teacher, this translates to:

$$\boldsymbol{p}_{i,t}^k = \mathtt{softmax}_L \left( h_t(\boldsymbol{c}_{i,t}^k)/\tau_t \right), \qquad i \in \{1, 2\} \qquad (5)$$

The student's projections are obtained analogously. The cross-view object-level loss is expressed as the averaged cross-entropy over all pairs of object representations and making the loss symmetric w.r.t. the student/teacher:

$$\mathcal{L}_{cv}^o = \frac{1}{K} \sum_{k \in [K]} \left( H(\boldsymbol{p}_{1,t}^k, \boldsymbol{p}_{2,s}^k) + H(\boldsymbol{p}_{2,t}^k, \boldsymbol{p}_{1,s}^k) \right) \qquad (6)$$

where $H(\mathbf{a}, \mathbf{b}) = -\sum_{l=1}^L \mathbf{a}_l \log(\mathbf{b}_l)$ denotes the cross-entropy.

**Cross-Image Object-Level Self-Supervision.** Once all object representations in the batch have been retrieved, the cross-image object-level loss is enforced in a manner analogous to its cross-view counterpart. The nearest neighbors are first fed to the projection head as follows:

$$\tilde{\boldsymbol{p}}_{i,t}^k = \mathtt{softmax}_L \left( h_t(\mathtt{nn}(\boldsymbol{c}_{i,t}^k))/\tau_t \right), \qquad i \in \{1, 2\} \qquad (7)$$

Note that these projections are only defined for the teacher since the $\mathtt{nn}(\cdot)$ operator is not differentiable. The formulation of the cross-image object-level loss aligns with that in Equation (6) up to the filtering of invalid positive pairs (see Sec. 3.2.2):

$$\mathcal{L}_{ci}^o = \frac{1}{Z_1} \sum_{k \in [K]} \mathbb{1}_{\{\mathcal{O}(\boldsymbol{c}_1^k)\}} H(\tilde{\boldsymbol{p}}_{1,t}^k, \boldsymbol{p}_{2,s}) + \frac{1}{Z_2} \sum_{k \in [K]} \mathbb{1}_{\{\mathcal{O}(\boldsymbol{c}_2^k)\}} H(\tilde{\boldsymbol{p}}_{2,t}^k, \boldsymbol{p}_{1,s}) \qquad (8)$$

where $\mathbb{1}_{\{\cdot\}}$ denotes the indicator function and where $Z_1$ and $Z_2$ are normalization constants *i.e.*, $\sum_{k \in [K]} \mathbb{1}_{\{\mathcal{O}(\boldsymbol{c}_1^k)\}}$ and $\sum_{k \in [K]} \mathbb{1}_{\{\mathcal{O}(\boldsymbol{c}_2^k)\}}$, respectively. Empirical evidence on the utility of the cycle consistency can be found in Appendix C.4.

**Global Cross-View Self-Supervision.** Importantly, both the clustering and the bootstrapping steps are contingent on the quality of the representations. To avoid potential instabilities, we incorporate a global cross-view loss into our framework, offering meaningful self-supervision at any stage of the pretraining. This loss is analogous to the one from Equation (6) except that it is applied to the global representation ($\boldsymbol{z}_{1,s}$, $\boldsymbol{z}_{2,s}$, $\boldsymbol{z}_{1,t}$ and $\boldsymbol{z}_{2,t}$), which are fed to a dedicated head $\bar{h}$:

$$\bar{\boldsymbol{p}}_{i,t} = \mathtt{softmax}_{\bar{L}} \left( \bar{h}_t(\bar{\boldsymbol{z}}_{i,t}/\bar{\tau}_t) \right), \qquad i \in \{1, 2\} \qquad (9)$$

Table 1: **Dense nearest neighbor retrieval**. The quality of the learned spatial features is probed with a k-NN classifier and using different ratios of training data. The depicted mIoU scores are derived from the validation sets of two scene-centric datasets. † refers to our own reproduction from official GitHub repositories. If not specified, publicly available checkpoints are used. ⋆ denotes results taken from Balažević et al. (2023).

| Method | Backbone | Params | Dataset | Epochs | ADE20K | | | | Pascal VOC | | | |
|---|---|---|---|---|---|---|---|---|---|---|---|---|
| | | | | | 1/128 | 1/64 | 1/8 | 1/1 | 1/128 | 1/64 | 1/8 | 1/1 |
| *Scene-centric* | | | | | | | | | | | | |
| SlotCon (Wen et al., 2022) | ResNet-50 | 25M | COCO | 800 | 9.9 | 11.8 | 17.3 | 22.1 | 37.3 | 42.8 | **52.9** | 57.2 |
| ORL (Xie et al., 2021a) | ResNet-50 | 25M | COCO | 800 | 9.1 | 10.3 | 13.6 | 16.6 | 30.2 | 33.3 | 42.0 | 45.6 |
| DINO† (Caron et al., 2021) | ViT-S/16 | 21M | COCO | 300 | 6.1 | 6.9 | 9.7 | 13.0 | 16.2 | 18.4 | 25.5 | 31.9 |
| MAE† (He et al., 2022) | ViT-S/16 | 21M | COCO | 300 | 3.7 | 4.1 | 5.4 | 6.8 | 8.5 | 9.3 | 12.2 | 15.9 |
| CrOC (Stegmüller et al., 2023) | ViT-S/16 | 21M | COCO | 300 | 7.6 | 9.0 | 13.1 | 18.0 | 27.1 | 31.4 | 40.3 | 47.1 |
| CrIBo | ViT-S/16 | 21M | COCO | 300 | **10.9** | **12.8** | **18.4** | **23.4** | **39.1** | **44.0** | 52.8 | **58.1** |
| *Object-centric* | | | | | | | | | | | | |
| SlotCon (Wen et al., 2022) | ResNet-50 | 25M | IN1K | 200 | 10.6 | 12.0 | 18.1 | 24.0 | 40.7 | 44.7 | 56.4 | 61.6 |
| DINO (Caron et al., 2021) | ViT-S/16 | 21M | IN1K | 800 | 9.4 | 10.6 | 14.6 | 18.4 | 24.5 | 28.7 | 38.7 | 46.1 |
| CrOC (Stegmüller et al., 2023) | ViT-S/16 | 21M | IN1K | 300 | 7.8 | 9.0 | 15.2 | 20.6 | 30.7 | 37.7 | 54.8 | 64.2 |
| TimeT (Salehi et al., 2023) | ViT-S/16 | 21M | IN1K+YTVOS | 800+30 | 11.7 | 13.6 | 19.7 | 24.6 | 38.9 | 44.3 | 56.0 | 62.7 |
| CrIBo | ViT-S/16 | 21M | IN1K | 800 | **13.7** | **16.5** | **23.2** | **28.3** | **52.7** | **59.3** | **69.3** | **73.2** |
| DINO (Caron et al., 2021) | ViT-B/16 | 85M | IN1K | 400 | 11.1 | 12.6 | 17.6 | 22.0 | 29.2 | 34.7 | 47.2 | 54.9 |
| MAE (He et al., 2022) | ViT-B/16 | 85M | IN1K | 1600 | 2.7 | 3.0 | 4.0 | 5.3 | 6.0 | 6.5 | 8.9 | 13.8 |
| LOCA⋆ (Caron et al., 2022) | ViT-B/16 | 85M | IN1K | 600 | - | - | - | 18.5 | - | - | - | 57.5 |
| Hummingbird⋆ (Balažević et al., 2023) | ViT-B/16 | 85M | IN1K | 300 | 11.7 | 15.1 | - | 28.3 | **50.5** | 57.2 | - | 70.5 |
| CrIBo | ViT-B/16 | 85M | IN1K | 400 | **13.2** | **16.5** | **23.6** | **30.0** | **50.5** | **60.3** | **70.8** | **74.9** |

where $\bar{\tau}_s$ and $\bar{\tau}_t$ are the corresponding temperature parameters for the $\bar{L}$-dimensional output distribution. The global cross-view loss is defined as:

$$\mathcal{L}_{cv}^g = H(\bar{\boldsymbol{p}}_{1,t}, \bar{\boldsymbol{p}}_{2,s}) + H(\bar{\boldsymbol{p}}_{2,t}, \bar{\boldsymbol{p}}_{1,s}) \tag{10}$$

where $H(\mathbf{a}, \mathbf{b}) = -\sum_{l=1}^{\bar{L}} \mathbf{a}_l \log(\mathbf{b}_l)$.

Finally, we obtain the overall training objective of CrIBo as a sum of its individual components:

$$\mathcal{L}_{\text{tot}} = \mathcal{L}_{cv}^g + \mathcal{L}_{cv}^o + \mathcal{L}_{ci}^o \tag{11}$$

An ablation study over the inclusion of the individual loss terms can be found in Table 5. As in Caron et al. (2021), the student's parameters are optimized to minimize the above loss whereas the parameters of the teacher are updated via an exponential moving average of the student's weights.

# 4 EXPERIMENTS

We first verify that the proposed pretraining method aligns well with the objective of in-context learning via nearest neighbor retrieval. We then show that in doing so, we do not compromise the performance on standard evaluations. General details on the experimental setup can be found in Appendix A.

## 4.1 DENSE NEAREST NEIGHBOR RETRIEVAL

Inspired by Balažević et al. (2023), we probe the quality of the learned features with a dense nearest neighbor (NN) classification task. The $k$-NN classifier is fitted on the local representations of a uniformly sub-sampled set of training images and evaluated on all the patches from the validation set of images. The sub-sampling factor is either 1, 8, 64 or 128. The only preprocessing applied to the images is a down-sampling to the closest resolution which is a multiple of the patch size. Following that step, patch-level labels are obtained via majority voting. We report the mIoU scores on Pascal VOC 2012 (Everingham et al.) and ADE20K (Zhou et al., 2017). When the sub-sampling ratio is greater than 1, the reported mIoU scores result from averaging over 5 independent runs. The number of nearest neighbor patches is set to $k = 50$.

In Table 1, we observe that using object-level nearest neighbor bootstrapping throughout training is an adequate pretext task for this evaluation, which is not a surprise due to the inherent similarities between the training and test phases. Overall, CrIBo outperforms all methods, but Hummingbird (Balažević et al., 2023), by a significant margin and particularly following the scene-centric

Table 2: **Linear segmentation with frozen backbones.** The linear decoder from Segmenter Strudel et al. (2021) is trained on the frozen spatial features obtained with various self-supervised learning methods. We report the mIoU scores achieved on the validation sets of 4 different datasets. † refers to our own reproduction from official GitHub repositories. If not specified, publicly available checkpoints are used.

| Method | Backbone | Params | Dataset | Epochs | Pascal Context | Pascal VOC | COCO-Stuff 164K | ADE20K |
|---|---|---|---|---|---|---|---|---|
| *Scene-centric* | | | | | | | | |
| DINO† (Caron et al., 2021) | ViT-S/16 | 21M | COCO | 300 | 27.3 | 43.9 | 19.9 | 18.9 |
| MAE† (He et al., 2022) | ViT-S/16 | 21M | COCO | 300 | 18.1 | 25.6 | 11.3 | 11.0 |
| CrOC (Stegmüller et al., 2023) | ViT-S/16 | 21M | COCO | 300 | 30.0 | 51.7 | 22.6 | 20.5 |
| CrIBo | ViT-S/16 | 21M | COCO | 300 | 37.5 | 61.0 | 29.1 | 27.5 |
| CrIBo | ViT-S/8 | 21M | COCO | 300 | **40.6** | **63.9** | **31.2** | **28.3** |
| *Object-centric* | | | | | | | | |
| DINO (Caron et al., 2021) | ViT-S/16 | 21M | IN1K | 800 | 32.4 | 49.0 | 24.1 | 23.4 |
| CrOC (Stegmüller et al., 2023) | ViT-S/16 | 21M | IN1K | 300 | 37.7 | 68.1 | 30.2 | 27.6 |
| TimeT (Salehi et al., 2023) | ViT-S/16 | 21M | IN1K+YTVOS | 800+30 | 40.6 | 67.9 | 32.0 | 29.9 |
| CrIBo | ViT-S/16 | 21M | IN1K | 800 | **41.7** | **73.7** | **33.8** | **31.9** |
| DINO (Caron et al., 2021) | ViT-B/16 | 85M | IN1K | 400 | 38.9 | 64.6 | 31.9 | 30.5 |
| MAE (He et al., 2022) | ViT-B/16 | 85M | IN1K | 1600 | 27.9 | 40.9 | 14.4 | 17.5 |
| CrIBo | ViT-B/16 | 85M | IN1K | 400 | **42.9** | **74.9** | **36.0** | **34.7** |

Table 3: **Finetuning evaluation with Segmenter.** Backbones pre-trained with different self-supervised learning methods are finetuned using Segmenter (Strudel et al., 2021). We report the mIoU scores achieved on the validation sets of 4 different datasets. † refers to our own reproduction from official GitHub repositories. If not specified, publicly available checkpoints are used.

| Method | Backbone | Params | Dataset | Epochs | Pascal Context | Pascal VOC | COCO-Stuff 164K | ADE20K |
|---|---|---|---|---|---|---|---|---|
| *Scene-centric* | | | | | | | | |
| DINO† (Caron et al., 2021) | ViT-S/16 | 21M | COCO | 300 | 33.5 | 66.1 | 35.6 | 35.0 |
| MAE† (He et al., 2022) | ViT-S/16 | 21M | COCO | 300 | 32.5 | 61.6 | 35.4 | 35.4 |
| CrOC (Stegmüller et al., 2023) | ViT-S/16 | 21M | COCO | 300 | 38.7 | 70.5 | 38.0 | 37.9 |
| CrIBo | ViT-S/16 | 21M | COCO | 300 | **41.9** | **74.2** | **39.4** | **39.3** |
| *Object-centric* | | | | | | | | |
| DINO (Caron et al., 2021) | ViT-S/16 | 21M | IN1K | 800 | 46.0 | 80.3 | 43.2 | 43.3 |
| CrOC (Stegmüller et al., 2023) | ViT-S/16 | 21M | IN1K | 300 | 46.0 | 80.9 | 42.9 | 42.8 |
| TimeT (Salehi et al., 2023) | ViT-S/16 | 21M | IN1K+YTVOS | 800+30 | 47.4 | 80.4 | 43.1 | 43.5 |
| CrIBo | ViT-S/16 | 21M | IN1K | 800 | **49.3** | **82.3** | **43.9** | **45.2** |
| DINO (Caron et al., 2021) | ViT-B/16 | 85M | IN1K | 400 | 45.8 | 82.2 | 44.4 | 45.0 |
| MAE (He et al., 2022) | ViT-B/16 | 85M | IN1K | 1600 | 47.9 | 82.7 | **45.5** | **46.4** |
| CrIBo | ViT-B/16 | 85M | IN1K | 400 | **49.2** | **83.4** | 44.6 | 46.0 |

pretraining. It is worth noting that the comparison with Balaževié et al. (2023) is not exactly *apple-to-apple*, as *i)* we do not use multiple augmentations per image and *ii)* we refrain from balancing the classes in the training set. These differences in the implementation of the evaluation reflect the divergence in our objectives: we place a higher emphasis on the assessment of the intrinsic quality of the learned representations rather than the absolute value of the reported results.

## 4.2 SEGMENTATION WITH A LINEAR HEAD

The frozen local representations are fed to the linear decoder head from Segmenter (Strudel et al., 2021) using the MMSegmentation (Contributors, 2020) implementation. Herein, spatial tokens undergo a linear projection to the class space, followed by bilinear up-sampling to match the input image dimensions, enabling the application of pixel-wise cross-entropy loss. Performance metrics, in terms of mIoU scores, are reported on the same 4 datasets (see Appendix A.2 for more details).

As reported in Table 2, CrIBo exhibits commendable performance in linear segmentation, following both scene-centric and object-centric pretrainings. Among the methods using a ViT-B/16, CrIBo strongly outperforms other baselines. In the case of ViT-S/16, CrIBo also outperforms all baselines but with a smaller margin, due mostly to TimeT (Salehi et al., 2023) being a particularly strong baseline. It is noteworthy that as opposed to prevailing studies, *e.g.*, Stegmüller et al. (2023); Bardes et al. (2022), we do not concatenate spatial tokens from multiple layers of the ViT. Indeed, this trick is typically used to compare on an equal footing with ResNet-50, which has a much larger output dimension. As we only use and compare with ViTs, we opt for evaluations incorporating as minimal parameters as possible, aspiring to present a more accurate reflection of the features' intrinsic quality.

Table 4: **Ablation study on the hyperparameters of CrIBo**. Here, a ViT-S/16 undergoes pretraining on COCO for 300 epochs with various combinations of hyperparameter values. The resulting features are then compared using dense nearest neighbors retrieval. We denote the size of the queues $\mathcal{M}_i$ in terms of the number of images present by $S$. Unless explicitly indicated, the values of the hyperparameters are set to $\lambda_{\text{pos}} = 2.0$, $S = 25\text{k}$, and $K = 64$.

| Dataset | $\mathcal{L}_{ci}^o$ | Positional weighting ($\lambda_{\text{pos}}$) | | | | | | Memory bank size ($S$) | | | Number of objects ($K$) | | | | |
|---|---|---|---|---|---|---|---|---|---|---|---|---|---|---|---|
| | | 0.0 | 0.1 | 1.0 | 2.0 | 4.0 | 8.0 | 1k | 5k | 25k | 4 | 8 | 16 | 32 | 64 |
| Pascal VOC | ✗ | 44.9 | 46.3 | 47.5 | 47.5 | 45.8 | 44.2 | - | - | - | 47.3 | 47.6 | 48.1 | 47.1 | 47.5 |
| Pascal VOC | ✓ | 55.9 | 57.1 | **58.1** | **58.1** | 57.5 | 57.0 | 46.7 | 47.3 | **58.1** | 49.9 | 52.9 | 54.6 | 57.0 | **58.1** |
| ADE20K | ✗ | 16.9 | 18.0 | 18.9 | 18.4 | 17.8 | 16.3 | - | - | - | 17.7 | 18.0 | 18.9 | 18.5 | 18.4 |
| ADE20K | ✓ | 21.6 | 21.8 | 23.1 | **23.4** | 22.6 | 22.5 | 18.1 | 18.4 | **23.4** | 18.1 | 19.4 | 20.3 | 21.7 | **23.4** |

### 4.3 END-TO-END FINETUNING WITH SEGMENTER

In a real-world scenario, self-supervised pretraining often precedes end-to-end finetuning on the target dataset with an application-specific decoder head. The ability of the different pretraining paradigms to cope with that setting is assessed with an end-to-end finetuning of the backbone topped with the transformer-based decoder from Segmenter (Strudel et al., 2021). Here, the backbone's spatial features are fed to a transformer encoder along with $K$ learnable class tokens. The resulting class and spatial tokens are projected onto one another to obtain patch-level predictions, subsequently upsampled to the input image's size to enforce the pixel-wise cross-entropy loss. We report the mIoU scores achieved on the same 4 datasets.

The insights gained from the preceding experiments (Tabs. 1 and 2) underscore that models pretrained with CrIBo have excellent generalists properties, *i.e.*, allowing them to perform various tasks without requiring finetuning. The results depicted in Table 3 indicate that, when required, these models can also evolve into remarkable specialists. All that aside, after fine-tuning, all models tend to perform comparably, highlighting that the finetuning may be a suboptimal regime to compare SSL pretraining methods.

### 4.4 ABLATIONS

We report in Table 4 the results of a grid search over various hyperparameters performed on COCO. Interestingly, we observe that CrIBo performs the best when operating in the overclustering regime, that is, with a $K$ much larger than the number of objects present in the images, but only when bootstrapping is used. We postulate that this is due to the fact that overclustering offers the possibility to find better matches. To illustrate, consider the image in Figure 2 depicting a zebra, a giraffe, and an elephant. With $K = 1$, this would necessitate identifying another image displaying the identical trio of animals; which is rare. With a larger value of $K$, the animals are isolated, expanding the number of good candidates. In the overclustering regime, the clusters represent object parts, which further facilitates the matchings. Similarly, a larger queue also amplifies the number of suitable candidates, a trend reflected in the results until reaching a saturation point at a queue size of 25k. More ablations are available in Appendix C.

## 5 CONCLUSION

We introduce CrIBo, the first end-to-end online cross-image object-level bootstrapping self-supervised learning method. CrIBo capitalizes on the augmented sample diversity derived from image-level bootstrapping while harnessing the advantages of localized self-supervision inherent to object-level learning. We extensively evaluate CrIBo across multiple dense downstream tasks showcasing excellent performance on in-context scene understanding tasks and highlighting the merits of the cross-image object-level learning paradigm.

**Limitations.** This study is developed around the ViT architecture as it has become quite ubiquitous. As such, and for the sake of coherence, we only compare with other ViT-based methods. As we have to cope with a computational budget, we chose to invest it in producing a well-grounded and motivated approach with extensive analysis, rather than in exploring larger datasets (Thomee et al., 2016; Changpinyo et al., 2021) or models, *e.g.*, ViT-Large or ViT-Huge.

**Acknowledgement.** This project is funded by the European Research Council (ERC) under the European Union's Horizon 2020 research and innovation program (Grant Agreement No. 101021347). This work is also partially funded by the Personalized Health and Related Technologies (PHRT), grant number 2021/344; as well as the Fonds Wetenschappelijk Onderzoek (FWO), project G0A4720N. We acknowledge both EuroCC Belgium and the Swiss National Supercomputing Centre (project ID 606) for awarding this project access to the LUMI supercomputer.

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

APPENDIX

# A   EXPERIMENTAL SETUP

## A.1   PRETRAINING

**Pre-training datasets.**   Our pretraining datasets include COCO (Lin et al., 2014) and ImageNet-1k (Deng et al., 2009). The former includes about 118k scence-centric images while the latter includes about 1.3M object-centric images. Both are ubiquitous in scene-centric self-supervised learning and object-centric self-supervised learning, respectively.

**Network architecture.**   We employ vision transformers as our backbone $f$. This selection aligns with its prevalent use in contemporary methodologies and allows for fair and easy comparisons with existing works. The configuration of the projection heads closely follows that of Caron et al. (2021). Both the image-level head $\bar{h}$ and object-level head $h$ share their weights, except for the final linear layer. Both the image-level and the object-level output distribution have the same size *i.e.*, $\bar{L} = L = 65,536$.

**Optimization.**   The ViT-small (ViT-S/16) is trained for 800 epochs, while the ViT-base (ViT-B/16) is trained for 400 epochs. The ViT-Base is only trained on ImageNet-1k, while the ViT-Small is trained on both COCO and ImageNet-1k. Pretrainings on COCO use a batchsize of 256 while pretrainings on ImageNet-1k use a batchsize of 1024. Learning rate, weight-decay, and other optimization-related hyperparameters are exactly the same as in DINO (Caron et al., 2021).

**Hyperparameters.**   Results reported in tables using ViT-S/16 (apart from the gridsearch) are based on the following hyperparameters: $(\lambda_{\text{pos}}, S, K) = (1.0, 25\text{k}, 32)$ and $(\lambda_{\text{pos}}, S, K) = (2.0, 25\text{k}, 64)$ for pretrainings on ImageNet-1K and COCO, respectively. Results reported for ViT-B/16 are based using the following hyperparameters: $(\lambda_{\text{pos}}, S, K) = (1.0, 25\text{k}, 32)$.

## A.2   EVALUATION PROTOCOLS

**Linear segmentation.**   The frozen local representations are fed to the linear decoder head from Segmenter (Strudel et al., 2021), and we use the implementation available through MMSegmentation (Contributors, 2020). Herein, spatial tokens undergo a linear projection to the class space, followed by bilinear up-sampling to match the input image dimensions, enabling the application of pixel-wise cross-entropy loss. Performance metrics, in terms of mIoU scores, are reported on four different datasets: Pascal Context (Mottaghi et al., 2014), Pascal VOC 2012 (Everingham et al.), COCO-Stuff 164K (Caesar et al., 2018) and ADE20K (Zhou et al., 2017).

Regardless of the dataset used, the crop size is always $512 \times 512$ pixels, and the dataset configurations are kept as provided by MMSegmentation (Contributors, 2020). For ADE20K and COCO-Stuff 164K, we use the *160k_iterations_schedule* whereas for Pascal VOC 2012 and Pascal Context we use the *80k_iterations_schedule*. The only change we make to the default schedules configurations is that we use the Adam optimizer (Kingma & Ba, 2014) instead of SGD. For each pretraining method and dataset, we run with four different learning rates (`8e-4`, `3e-4`, `1e-4`, and `8e-5`) and report the highest mIoU score.

**Finetuning with Segmenter.**   We perform an end-to-end finetuning of the backbone topped with the transformer-based decoder from Segmenter (Strudel et al., 2021). Here, the backbone's spatial features are fed to a transformer encoder along with $K$ learnable class tokens. The resulting class and spatial tokens are projected onto one another to obtain patch-level predictions and subsequently up-sampled to the input image's size to enforce the pixel-wise cross-entropy loss. We report the mIoU scores achieved on 4 different datasets: Pascal Context (Mottaghi et al., 2014), Pascal VOC 2012 (Everingham et al.), COCO-Stuff 164K (Caesar et al., 2018) and ADE20K (Zhou et al., 2017).

Regardless of the dataset used, the crop size is always $512 \times 512$ pixels, and the dataset configurations are kept as provided by MMSegmentation (Contributors, 2020). For ADE20K and COCO-Stuff 164K, we use the *160k_iterations_schedule* whereas for Pascal VOC 2012 and Pascal Context,

we use the *80k_iterations_schedule*. Two changes are made w.r.t. the default schedules configurations. First, we use the Adam optimizer (Kingma & Ba, 2014) instead of SGD and second, we use `eta_min = 0.1 · lr` instead of `eta_min = 1e-4`. For each pretraining method and dataset, we run with four different learning rates (`8e-5`, `3e-5`, `1e-5`, and `8e-6`) and report the highest mIoU score.

## B    CLUSTERING ALGORITHM

We briefly detail the clustering algorithm, particularly the distinctions from its original implementation proposed in Stegmüller et al. (2023).

The overall objective of the clustering step is to assign the spatial tokens from **both** views of a common image to $K$ semantically coherent groups. Formally, given the dense representations $z_1$ and $z_2$ of two augmented views, the joint representation $z_{\text{cat}} \in \mathbb{R}^{2N \times d}$ is obtained by a concatenation along the dimension of the tokens. The $k^{th}$ centroid $c^k \in \mathbb{R}^d$ is initialized by uniformly sampling without replacement one of the $2N$ tokens. The clustering procedure relies on the Sinkhorn-Knopp algorithm (Cuturi, 2013) to solve the optimal transportation problem of assigning $2N$ tokens to $K$ clusters/centroids. The transportation cost between tokens and centroids $\mathbf{T}^{(\text{sem})} \in \mathbb{R}^{2N \times K}$ is given by the negative cosine distance:

$$\mathbf{T}^{(\text{sem})}_{n,k} = -\frac{< c^k, z^n_{\text{cat}} >}{\|c^k\| \, \|z^n_{\text{cat}}\|} \tag{12}$$

The Sinkhorn-Knopp algorithm efficiently computes a solution (an assignment matrix $\mathbf{Q}^*$) to the following optimization problem:

$$\mathbf{Q}^* = \underset{\mathbf{Q} \in \mathcal{Q}}{\text{argmin}} < \mathbf{Q}, \mathbf{T}^{(\text{sem})} > -\frac{1}{\lambda} H(\mathbf{Q}) \tag{13}$$

which aims to minimize the inner product of the transportation plan and the transportation cost, subject to an entropy constraint on the assignments. The transportation polytope $\mathcal{Q}$ defines the set of valid assignments:

$$\mathcal{Q} = \{\mathbf{Q} \in \mathbb{R}^{2N \times K}_+ \mid \mathbf{Q}\mathbf{1}_K = \frac{1}{2N}\mathbf{1}_{2N}, \mathbf{Q}^\top\mathbf{1}_{2N} = \frac{1}{K}\mathbf{1}_K\} \tag{14}$$

We update the centroids by pooling over each cluster:

$$c^k = \sum_{n \in [2N]} \mathbf{Q}^*(n, k) \cdot z^n_{\text{cat}} \tag{15}$$

At this point, we could stop, but we observed improved results when doing multiple iterations of the above steps, *i.e.*, restarting the procedure from Equation (12) and the centroids found in Equation (15). Empirically, we found that 5 iterations is a good trade-off between speed and accuracy. In practice, we use the "hard" version of the assignments obtained by ensuring that each token is only assigned to a single cluster. After a final $l_1$-normalization of the columns of the herein obtained matrix, the view-wise object representations are computed as in Equation (1).

Importantly, the above-defined algorithm diverges from the one proposed in CrOC (Stegmüller et al., 2023) in multiple ways. The main difference is that we do not rely on an iterative procedure to find the optimal number of centroids $K$, and as such, the number of clusters is fixed. Another discrepancy is that we use uniform distributions for both the sampling of the initial centroids and the marginals of the assignments (see Eqs. (13) and (14)). We observed that these modifications simplify the overall algorithm without incurring a drop in performance. In the following paragraph, we discuss our adjustments to the positional cues of the clustering algorithm.

**Positional cues**    An important property of the selected clustering algorithm is its ability to leverage positional cues to compensate for the lack of high-level semantics in the features at the beginning of the training. The spatial guidance is incorporated in the algorithm via an additive constraint on the transportation cost matrix:

$$\mathbf{T}^{(\text{tot})} = \mathbf{T}^{(\text{sem})} + \lambda_{\text{pos}}\mathbf{T}^{(\text{pos})} \tag{16}$$

Table 5: **Ablation study on the individual contribution of each loss.** ViT-S/16 are trained on COCO for 300 epochs and evaluated with downstream tasks of different levels of granularity.

| $\mathcal{L}_{cv}^{g}$ | $\mathcal{L}_{cv}^{o}$ | $\mathcal{L}_{ci}^{o}$ | Patch-level NN | | Object Detection with YOLO-S | | | Image-level NN |
|---|---|---|---|---|---|---|---|---|
| | | | Pascal VOC (mIoU) | ADE20k (mIoU) | COCO (AP$^{bb}$) | COCO (AP$^{bb}_{50}$) | COCO (AP$^{bb}_{75}$) | IN1K (top-1 Acc.) |
| ✓ | ✗ | ✗ | 31.6 | 12.8 | 24.0 | 40.6 | 23.9 | 34.3 |
| ✓ | ✓ | ✗ | 47.1 | 18.3 | 27.1 | 44.2 | 27.6 | 35.8 |
| ✓ | ✓ | ✓ | **58.1** | **23.4** | **30.2** | **48.4** | **31.0** | **38.2** |

Table 6: **Object detection with YOLO-S.** Backbones pre-trained with different self-supervised learning methods are finetuned with YOLO-S (He et al., 2017). We report the mean average precision for the predicted bounding boxes (AP$^{bb}$) for different IoU thresholds on COCO.

| Method | Backbone | Params | Dataset | Epochs | AP$^{bb}$ | AP$^{bb}_{50}$ | AP$^{bb}_{75}$ |
|---|---|---|---|---|---|---|---|
| *Object-centric* | | | | | | | |
| DINO Caron et al. (2021) | ViT-S/16 | 21M | IN1K | 800 | 35.4 | **56.0** | 36.6 |
| CrOC Stegmüller et al. (2023) | ViT-S/16 | 21M | IN1K | 300 | **35.7** | 55.5 | 37.0 |
| TimeT Salehi et al. (2023) | ViT-S/16 | 21M | IN1K+YTVOS | 800+30 | 35.0 | 55.5 | 36.2 |
| *Ours* | | | | | | | |
| CrIBo | ViT-S/16 | 21M | IN1K | 800 | **35.7** | 55.6 | **37.4** |

Similar to CrOC, we use the 2D coordinates $\mathbf{E}^{(\text{cat})} \in \mathbb{R}^{2N \times 2}$ of every patch/token in the joint dense representation $z_{\text{cat}}$ w.r.t. the upper left corner of the image that generated the two views. Up to normalization constants, the distance between the $i^{th}$ token and the $j^{th}$ centroid is given by:

$$\mathbf{T}_{ij}^{(\text{pos})} = \arg \min_{l \in \mathcal{S}_j} ||\mathbf{E}_i^{(\text{cat})} - \mathbf{E}_l^{(\text{cat})}||_2 \tag{17}$$

where $\mathcal{S}_j$ is the set of tokens assigned to the $j^{th}$ centroid from the previous iteration. It is noteworthy that the positional cost differs from that of CrOC, which sets the position of a given cluster as the average position of the tokens assigned to it. In our approach, the distance between a token and a given centroid depends on the query token. This mitigates the issue of concentric objects sharing the same position.

In Table 4, we ablate over the hyperparameter $\lambda_{\text{pos}}$, which modulates the importance of the positional cost in the clustering algorithm.

## C ADDITIONAL EXPERIMENTS

### C.1 CONTRIBUTION OF INDIVIDUAL LOSS TERMS

Consecutive evaluations showcase the contribution of each loss in Table 5 on 3 different downstream tasks. It follows that all losses contribute a significant improvement to the overall performance of CrIBo.

### C.2 OBJECT DETECTION WITH YOLO-S

We consider an object detection downstream task to shed a different light on the quality of the learned embeddings. We perform an end-to-end finetuning of the backbone using YOLO-S (Fang et al., 2021) on COCO Lin et al. (2014). We assess performance based on the standard Average Precision metrics, namely *i.e.*, AP$^{bb}_{50}$, AP$^{bb}_{75}$ and AP$^{bb}$, computed from bounding boxes predicted on the validation set. We strictly follow the finetuning protocol from Fang et al. (2021), with the sole modification being an adjustment to a batchsize of 32, as opposed to the originally used batchsize of 8.

Our choice of object detection method was guided by its simplicity and minimal requirement for additional parameters, aiming to capture the nuances of various pretraining methods. However, Table 6 reveals that YOLO-S (Fang et al., 2021) is relatively insensitive to the pretraining scheme and provides less information than anticipated. In hindsight, it might have been more judicious to start

query | clustering | object-level cross-image matches

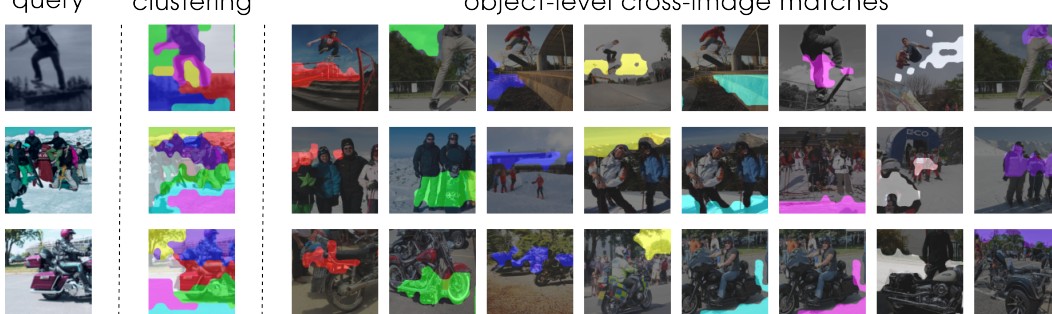

Figure 4: **Visualization of cross-image object-level matchings on COCO.** For a given query view (considered as view 1 here), object representations are computed via clustering. For each object-representation $c_1^k$ (highlighted in unique colors), its nearest neighbor $\text{nn}(c_1^k, \mathcal{M}_1)$ in the memory bank $\mathcal{M}_1$ is visualized. In this visualization, $K = 12$.

with the parameters specific to YOLO-S already trained. Indeed, this approach involves appending a hundred detection tokens to the image tokens before transmitting the resulting sequence through the transformer encoder. As such, earlier layers are at risk of receiving strong and biased gradients at the beginning of the finetuning phase, potentially overriding the properties learned during pretraining and explaining the observed results. On the other hand, the results in Table 6 are typically in the ballpark of those reported in the original paper (Fang et al., 2021).

### C.3 SEGMENTATION WITH A LINEAR HEAD AND CONCATENATION

We replicate the same linear segmentation experiment from the main text (see Section 4.2), but we concatenate the spatial tokens from the last 4 layers of the ViTs in Table 7. This scenario yields improved overall performance, but it's not clear whether these improvements result from the fact that different layers of the ViT encode different information or from the increased number of trainable parameters.

### C.4 CYCLE CONSISTENCY BRINGS STABILITY

We evaluate the effect of the cycle consistency criterion by comparing the results of various pretraining settings with and without the bootstrapping condition. Training is performed on ImageNet-1k for 300 epochs for every combination of the following hyperparameters:

- $K$: $\{4, 8, 12\}$.
- $\lambda_{\text{pos}}$: $\{1., 2., 4.\}$.

Table 7: **Linear segmentation with frozen backbones.** The linear decoder from Segmenter (Strudel et al., 2021) is trained on the frozen and concatenated spatial features from the last 4 layers of ViTs pretrained with various self-supervised learning methods. We report the mIoU scores achieved on the validation sets of 4 different datasets. Publicly available checkpoints are used when possible, and our reproduced results are indicated with a † symbol.

| Method | Backbone | Params | Epochs | Dataset | Pascal Context | Pascal VOC | COCO-Stuff 164K | ADE20K |
|---|---|---|---|---|---|---|---|---|
| *Scene-centric* | | | | | | | | |
| DINO† (Caron et al., 2021) | ViT-S/16 | 21M | 300 | COCO | 28.9 | 47.4 | 21.6 | 20.3 |
| MAE† (He et al., 2022) | ViT-S/16 | 21M | 300 | COCO | 19.9 | 29.4 | 13.0 | 13.5 |
| CrOC (Stegmüller et al., 2023) | ViT-S/16 | 21M | 300 | COCO | 32.2 | 54.7 | 25.9 | 24.6 |
| CrIBo | ViT-S/16 | 21M | 300 | COCO | **37.9** | **62.4** | **30.3** | **29.9** |
| *Object-centric* | | | | | | | | |
| DINO (Caron et al., 2021) | ViT-S/16 | 21M | 800 | IN1K | 41.3 | 69.0 | 33.0 | 31.0 |
| CrOC (Stegmüller et al., 2023) | ViT-S/16 | 21M | 300 | IN1K | 39.9 | 71.4 | 33.4 | 31.3 |
| TimeT (Salehi et al., 2023) | ViT-S/16 | 21M | 800+30 | IN1K+YTVOS | 41.9 | 71.9 | 34.5 | 32.3 |
| CrIBo | ViT-S/16 | 21M | 800 | IN1K | **42.4** | **75.2** | **34.6** | **33.4** |
| DINO (Caron et al., 2021) | ViT-B/16 | 85M | 400 | IN1K | **43.1** | 74.2 | 29.9 | 34.5 |
| MAE (He et al., 2022) | ViT-B/16 | 85M | 1600 | IN1K | 32.7 | 53.0 | 19.7 | 23.5 |
| CrIBo | ViT-B/16 | 85M | 400 | IN1K | 43.0 | **75.7** | **36.5** | **35.4** |

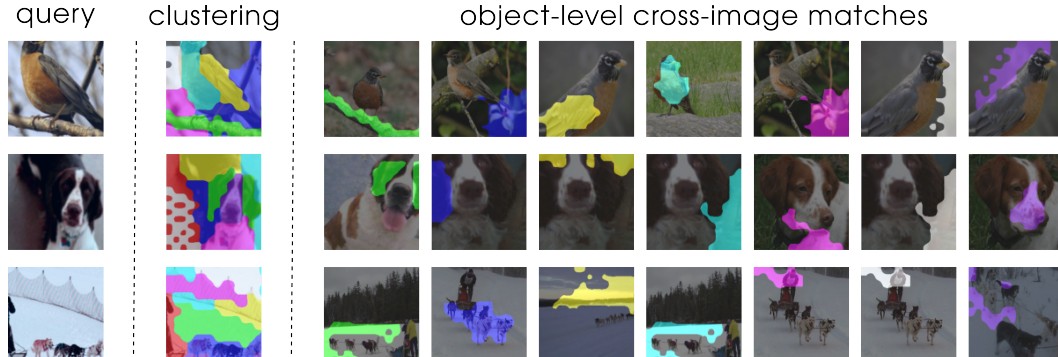

Figure 5: **Visualization of cross-image object-level matchings on ImageNet-1k.** For a given a query view (considered as view 1 here), object representations are computed via clustering. For each object-representation $c_1^k$ (highlighted in unique colors), its nearest neighbor $\text{nn}(c_1^k, \mathcal{M}_1)$ in the memory bank $\mathcal{M}_1$ is visualized. In this visualization, $K = 12$.

- $S$: $\{25\text{k}, 50\text{k}\}$.

We then evaluate the resulting models with the dense nearest neighbor retrieval task (see Section 4.1). In Table 8, we report the aggregated mIoU results for two datasets (Pascal VOC 2012 (Everingham et al.) and ADE20K (Zhou et al., 2017)) and the two settings, *i.e.*, with and without the cycle consistency condition. The aggregation is either the minimum, the maximum, or the average of all the results. As can be observed in Table 8, the proposed criterion does not bring improvements *per se*, but stabilizes the training procedure. Indeed, under the systematic bootstrapping regime, the performance can be unpredictable, as exemplified by the large variations of mIoU scores. This happens when nearest neighbors are determined based on low-level cues, a scenario that adaptive bootstrapping strives to prevent through the employment of cycle consistency.

## C.5 VISUALIZATION OF CROSS-IMAGE OBJECT-LEVEL MATCHINGS

Figure 4 and Figure 5 show the cross-image object-level matchings during a pretraining on COCO and ImageNet-1k, respectively. Even though ImageNet-1k is an object-centric dataset, classes can always be decomposed into finer grained classes. For example, the first image with the bird has tree branches present, which CrIBo does not fail to recognize.

## C.6 BOOTSTRAPPING RATIO

The evolution of the bootstrapping ratio over the training epochs is illustrated in Figure 6. Using the notations from the paper, the bootstrapping ratio corresponds to approximately $\frac{Z_1+Z_2}{2K}$ averaged over the whole dataset. $Z_1$ and $Z_2$ are the normalization constants from Equation (8). In practice, the number of object representations in an image is slightly smaller than $K$ as we discard the ones that do not span both views. At the initial epoch, the encoders are randomly initialized, making it very hard for the object representations to satisfy the cycle consistency condition. However, as learning progresses, representations become more semantically driven, and more bootstrapping is allowed. It

Table 8: **Ablation study on the adaptive bootstrapping using the cycle consistency condition.** Results are aggregated over multiple hyperparameters. Trained on ImageNet-1k for 300 epochs and evaluated using dense nearest neighbor retrieval (Tab. 1).

| Bootstrap criterion | Pascal VOC | | | ADE20k | | |
|---|---|---|---|---|---|---|
| | min. mIoU | max. mIoU | avg. mIoU | min. mIoU | max. mIoU | avg. mIoU |
| None | 49.1 | 71.2 | 68.8 | 16.4 | 27.2 | 25.7 |
| Cycle-consistency | **68.0** | **71.5** | **69.9** | **24.8** | **27.6** | **26.0** |

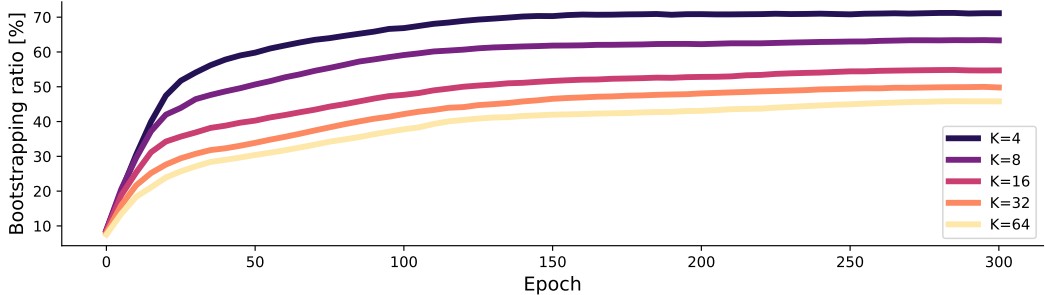

Figure 6: **Bootstrapping ratio over the epochs.** More bootstrapping is allowed as the training progresses, and as the representations become more semantically driven.

Table 9: **High-level profiling of CrIBo**. Experiments are run on a single node with 4x AMD MI250x (2 compute die per GPU *i.e.*, `worldsize = 8`) with a memory usage of 43.5 GB per compute die. The backbone is a ViT-B/16 and the batchsize is set to 128 per compute die *i.e.*, 1024 in total and $K = 12$. CrIBo specific operations are highlighted (operations specific to object-level nearest neighbors).

| Description | Operation | Absolute time per iteration [ms] | Relative time [%] |
|---|---|---|---|
| Forward pass | $f_t(\cdot) + f_s(\cdot) + \bar{h}_t(\cdot) + \bar{h}_s(\cdot)$ | 453.4 | 41.7 |
| Clustering | $\mathbf{Q}^* = \mathcal{C}(\cdot)$ | 82.2 | 7.6 |
| NN retrieval | $\texttt{nn}(\cdot) + \texttt{update}(\mathcal{M}_i)$ | 47.5 | 4.4 |
| Object projections | $h_t(\cdot) + h_s(\cdot)$ | 11.2 | 1.0 |
| Weights update | `backprop.` + EMA | 493.1 | 45.3 |
| *total* | | 1087.4 | 100 |

is worth noting that a higher $K$ results in a lower bootstrapping ratio as both the batch and the queue are larger (in terms of object representations), which imposes a more stringent condition to meet.

### C.7  HIGH-LEVEL PROFILING

A detailed timing analysis of CrIBo is performed in Table 9. It is worth noting that CriBo-specific operations take less than 15% of the total time.

### C.8  RUNTIME COMPARISON

A runtime comparison with other methods can be found in Table 10. CrIBo is faster than DINO and slower than MAE. Note that the quantitative comparisons in our paper are done with MAE trained for 1600 epochs and DINO/CrIBo trained for 400 epochs. This leads to a similar total computational budget.

Table 10: **Runtime comparison.** Experiments are run on a single node with 4x AMD MI250x (2 compute die per GPU i.e. `worldsize = 8`).

| Method | Time per epoch [minutes:seconds] | Memory per compute die [GB] | Batchsize |
|---|---|---|---|
| MAE | 04:45 | $\sim 36$ | 4096 |
| DINO | 29:43 | $\sim 45$ | 1024 |
| CrIBo ($K = 12$) | 23:58 | $\sim 44$ | 1024 |
| CrIBo ($K = 32$) | 28:22 | $\sim 60$ | 1024 |

Table 11: **We investigate the usage of supervision in CrIBo.** To get a sense of how supervision could help CrIBo, we ablate over the usage of labels for the clustering step and for the bootstrapping criterion. The hyperparameters (where applicable) are set to $(\lambda_{\text{pos}}, S, K) = (2.0, 25\text{k}, 64)$.

| Supervision | | Patch-level NN | |
|:---:|:---:|:---:|:---:|
| Clustering | Bootstrapping | Pascal VOC | ADE20k |
| ✓ | ✓ | 52.5 | 21.2 |
| ✓ | ✗ | 52.3 | 20.2 |
| ✗ | ✓ | **59.5** | **23.4** |
| ✗ | ✗ | 58.1 | **23.4** |

### C.9  SUPERVISED ORACLES

An ablation using different supervised oracles is shown in Table 11. The supervision takes place at two different levels: 1) the clustering and 2) the bootstrapping criterion.

1. Instead of using an unsupervised clustering algorithm for computing object representations, the semantic segmentation labels from COCO and COCO-Stuff are used. The masks are downsampled (16x) to match the dense representation output resolution. Given an image, the object representation associated with a given mask label is the average pooling of the patches assigned to the label.

2. Instead of using the cycle-consistency condition for discarding supposed invalid pairs of NNs, the downsampled (16x) semantic masks of COCO and COCO-Stuff are used. A pair of NN object representations is deemed valid if the union of all patch labels is the same in both object representations.

Surprisingly, using unsupervised clustering results in better performance than using the labels from COCO. In hindsight, this confirms again that the overclustering regime is where CrIBo is most at ease. Indeed, the granularity of the mask annotations in COCO is not particularly high. The results would have probably been different using finer-grained annotations. However, replacing the cycle-consistency condition with a supervised oracle yields modest improvements.

### C.10  CLUSTERING ALGORITHM

The clustering algorithm plays a pivotal role in the overall performance of CrIBo. We compare the clustering algorithm, detailed in Appendix B, with a well-established alternative, *i.e.*, the K-Means algorithm. Consequently, we pretrain ViT-S/16 for 300 epochs on COCO (Lin et al., 2014), using a memory bank of size $S = 25\text{k}$, various numbers of clusters $K$ along with the K-Means algorithm.

Table 12 depicts the performance achieved by each clustering algorithm on the dense nearest neighbor retrieval task (see Sec. 4.1). It can be observed that using the Sinkhorn-based clustering from Appendix B leads to an improvement over the K-Means algorithm. This difference is only partially explained by the lack of positional cues in K-Means as shown by the row $\lambda_{\text{pos}} = 0.0$. Indeed, the Sinkhorn-based method, deprived of positional cues, still yields better scores, confirming the utility of the non-standard clustering algorithm.

Table 12: Ablation study on the clustering algorithm.

| Clustering | $K = 4$ | | $K = 8$ | | $K = 16$ | | $K = 32$ | | $K = 64$ | |
|:---|:---:|:---:|:---:|:---:|:---:|:---:|:---:|:---:|:---:|:---:|
| | Pascal VOC | ADE20K | Pascal VOC | ADE20K | Pascal VOC | ADE20K | Pascal VOC | ADE20K | Pascal VOC | ADE20K |
| K-Means | 42.8 | 12.5 | 47.6 | 13.8 | 49.1 | 14.3 | 52.0 | 15.3 | 53.0 | 16.0 |
| Sinkhorn ($\lambda_{\text{pos}} = 0.0$) | 47.2 | 16.7 | 50.1 | 17.7 | 52.4 | 18.9 | 53.8 | 20.1 | 55.9 | 21.6 |
| Sinkhorn ($\lambda_{\text{pos}} = 2.0$) | **49.9** | **18.1** | **52.9** | **19.4** | **54.6** | **20.3** | **57.0** | **21.7** | **58.1** | **23.4** |

# D  DATASETS

**Pascal VOC 2012 (Everingham et al.)**    A dataset composed of a training set includes 10,582 images distributed across 21 classes, with one being a background class. The validation set incorporates 1,449 images.

**Pascal Context (Mottaghi et al., 2014)**    A scene-centric dataset comprises 4,998 training images spanning 60 semantic classes (including the background). The validation set contains 5,105 images.

**COCO-Stuff 164K (Caesar et al., 2018)**    A scene-understanding dataset, featuring labels spanning 91 "stuff" categories and 80 "things" categories. The training set is composed of 118K images, while the validation set contains 5K images.

**ADE20K (Zhou et al., 2017)**    A dataset that encompasses scenes featuring fine-grained labels across 150 distinct semantic categories, making it one of the most demanding available semantic segmentation datasets. The training set comprises 20,210 images, and the validation set consists of 2,000 images.

