# OpenReview forum: "CrIBo: Self-Supervised Learning via Cross-Image Object-Level Bootstrapping"
_ICLR.cc/2024/Conference — ICLR 2024 spotlight_

### Official Review · Reviewer_4jZr · 2023-10-20

**Soundness:** 3 good
**Presentation:** 3 good
**Contribution:** 3 good
**Rating:** 8
**Confidence:** 5

**Summary:**

The authors propose a self-supervised cross-image object-level bootstrapping method for dense visual representation learning. Specifically, the authors first use clustering algorithms to cluster dense features to obtain object features. Afterwards, the authors use object features from one view to retrieve object-level nearest neighbors from the memory bank, and then enforce consistency between each retrieved neighbor and its corresponding object feature from another view. Extensive experiments on multiple dense downstream tasks demonstrate the superiority of the proposed cross-image object-level learning paradigm.

**Strengths:**

-	The paper is well-motivated. The authors focus on two kinds of self-supervision in SSL: (i) cross-image self-supervision, and (ii) object-level self-supervision. Compared with cross-view self-supervision, cross-image self-supervision can provide more natural and diverse inter-image invariance. Compared with image-level self-supervision, object-level self-supervision is finer-grained and more suitable for scene-centric datasets. Based on this motivation, the authors extend existing cross-image SSL from the image level to the object level.

-	The paper is generally well-written and easy to follow.

-	The authors conduct extensive experiments and demonstrate promising results on various dense downstream tasks.

**Weaknesses:**

-	The authors claim that CrIBo is the first SSL approach that explicitly enforces cross-image consistency at the object/object-part level. However, some prior SSL works (e.g., [1]) have already performed cross-image SSL at the object/object-part level. Specifically, [1] enforces cross-image object-level consistency with KNN-retrieved cross-image object-instance pairs. The main difference is that [1] uses region proposal algorithms to produce region-level object features, whereas CrIBo uses clustering algorithms to produce pixel-level object features. Therefore, I suggest the authors discuss the differences with [1] and modify the claim accordingly.

-	For cross-image self-supervision, there are several closely related works (e.g., [2, 3, 4]) that should also be discussed in Sec. 2.

-	It seems that CrIBo does not perform very well in the fine-tuning evaluation. For example, as shown in Table 3, CrIBo performs worse than MAE when pre-trained on IN1K. Using a larger ViT-B/16 backbone even degrades the performance. Could the authors provide some explanations on this?

-	The authors use the clustering algorithm in [5]. Could the authors provide the justification on this? What if other clustering algorithms (e.g., k-means) are used? Since the quality of object representations depends on the clustering algorithms, an ablation study on different clustering algorithms would be an interesting experiment to explore.

-	It seems that CrIBo tends to induce heavy computational costs. A computational cost comparison (e.g., training time, GPU memory) with previous methods is preferred.

**Questions:**

I have some additional questions and suggestions:

-	The authors use the clustering algorithm to group dense features in the feature space to obtain object representations. What if using some heuristic algorithms (e.g., Multiscale Combinatorial Grouping [6]) to directly produce object masks on the input images first and then obtaining the corresponding object representations in the feature space (like what is done in DetCon [7]). Will the grouping in the feature space be better than the grouping in the image space? Furthermore, the authors may also want to consider using ground-truth object masks (e.g., mask annotations from COCO) to replace the clustering algorithm to see whether the supervised object annotations are the upper bound of the proposed method or the unsupervised grouping could even exceed the supervised counterpart. These experiments can add more insights to the paper.

-	According to Table 4, using a large memory bank size tends to improve the performance. What if using a memory bank size larger than 25k? Given the current ablation results, it seems that the performance does not saturate and may be further improved with a larger memory bank size.

-	Apart from ViT, is CrIBo applicable to other backbone architectures (e.g., CNN)?


**References:**

[1] Unsupervised Object-Level Representation Learning from Scene Images. In NeurIPS, 2021.

[2] Local Aggregation for Unsupervised Learning of Visual Embeddings. In ICCV, 2019.

[3] Delving into Inter-Image Invariance for Unsupervised Visual Representations. In IJCV, 2022.

[4] Mine Your Own View: Self-Supervised Learning Through Across-Sample Prediction. In arXiv, 2021.

[5] CrOC: Cross-View Online Clustering for Dense Visual Representation Learning. In CVPR, 2023.

[6] Multiscale Combinatorial Grouping. In CVPR, 2014.

[7] Efficient Visual Pretraining with Contrastive Detection. In ICCV, 2021.

---

> ### Author Response · Authors · 2023-11-21
>
> ### Thank you for the time taken to review our paper and the insightful feedback. Below, we answer all of your questions / concerns.
>
> > First SSL approach that explicitly enforces cross-image consistency at the object/object-part level claim
>
> Thank you for pointing this out. We have modified our claim accordingly. There are two main differences between ORL and CrIBo. Firstly, unlike the three-stage approach of ORL, CrIBo is a single-stage procedure where cross-image matches are continuously found throughout training. Secondly, CrIBo uses mask-based pooling for object-level representations, aligning it more with segmentation tasks, whereas ORL relies on bounding box-level pooling, making it more adequate for detection tasks.
>
> > Other related works
>
> These works are indeed relevant and have been added to the "Related Works" section of the revised draft.
>
> > Underperforming ViT-B in the fine-tuning evaluation?
>
> We have identified two primary factors contributing to this outcome.
>
> Firstly, owing to constraints in time and computational resources, our hyperparameter grid search was exclusively conducted for the ViT-S/16 on ImageNet-1k. The best-performing hyperparameters from this search were then applied to both the ViT-S/16 and the ViT-B/16, creating an inherent bias in favor of the former. We have now conducted an independent grid search for the ViT-B/16 and report improved scores, which are detailed in the revised manuscript.
>
> Secondly, we posit that the number of pretraining epochs significantly influences the downstream performance of this task. To support this claim, we report  the performance achieved by a ViT-S/16 pretrained for 300 epochs with CrIBo. These additional results underscore the advantage of extended training schedules for optimizing the performance in this task.
>
> > Supervised and image-space clustering
>
> We have conducted the requested experiment, i.e. using mask annotations from COCO instead of the clustering algorithm (see Appendix B.8 of the revised draft). Surprisingly, using unsupervised clustering results in better performance than using the labels from COCO. In Table 10, we also report minor improvements in using labels to determine when to enforce consistency between pairs of nearest neighbors.
>
> We did not experiment with clustering algorithms in the image space. However, a desired property of the clustering is that it groups semantically related patches together. If the pretraining is such that downstream tasks are more easily solved using the latent space rather than the raw image space (i.e. the latent space better captures the semantics), it also makes sense to cluster in the latent space to capitalize on the learning.
>
> > Larger memory bank?
>
> Our grid search for the memory bank size was initially set up with 1k, 5k, 25k, and 125k, scaling up by a factor of five each time. With a queue size of 125k, we ran into out-of-memory exceptions, so we only included results for 1k, 5k, and 25k. However, we then tried a queue size of 50k, which gave results similar to the 25k. We opted not to include it in the table primarily because it breaks the 5x increase pattern we were following. However, we have discussed these findings in Section 4.4 of our manuscript, mentioning that the performance gains appear to plateau at a 25k memory bank size.
>
> > Compatibility of CrIBo with other backbones
>
> In theory, there is no conceptual obstacle to the applicability of CrIBo to other architectures, e.g., convolutional neural networks. However, transformer-based architectures possess certain inherent features that make them well-suited for CrIBo. These architectures are designed to explicitly leverage inter-tokens similarities and (soft) pooling mechanisms.

---

> > ### Comment · Reviewer_4jZr · 2023-11-23
> > **Official Comment by Reviewer 4jZr**
> >
> > Thanks authors for the response. Most of my concerns are addressed while some remain. First, I appreciate that the authors provided the supervised oracle experiments in Appendix B.8. However, it is a pity that the authors did not provide any ablation study on different clustering algorithms. Besides, I would like to see the computational cost comparison with previous methods. However, it seems that the authors only provided the profiling of CrIBo in Table 9. These experiments can add more insights to the paper, but it is a pity that I do not see them in the rebuttal. I understand that the limited time and computational resources may prevent some of these experiments. Overall, I think this paper is worth sharing with the community. I would like to raise my score to 8 to reflect the efforts the authors have made during the rebuttal. I highly recommend the authors incorporate all the comments as discussed and provide the remaining experiments in the final version.

---

> ### Author Response · Authors · 2023-11-23
>
> Thank you for quick response giving us one final chance to reply!
>
> > Different clustering
>
> Given the time constraint of the discussion period and the many experiments asked by other reviewers, we had to prioritize some experiments. Indeed, $\lambda_{\text{pos}}$ is a hyperparameter specific to the clustering and varying its value is already an ablation on the clustering (see Table 4).
>
> Nonetheless, we'll provide a comparison with the ubiquitous K-Means algorithm on COCO for the camera-ready version.
>
> > Cost comparison with other methods
>
> We agree that such a comparison would be useful for the community. However, this requires installing dependencies and running code from numerous online repositories, which is not trivial considering that our platform is AMD/ROCm-based.
>
> However, since R3 is also keen on seeing this, we agree to provide such a table for a limited number of noteworthy methods e.g. DINO and MAE.
>
> As a side note, our implementation is based on DINO, which discloses its runtimes, thus CrIBo's runtimes are expected to be 1.15x slower than that.

---

### Official Review · Reviewer_w5sb · 2023-10-29

**Soundness:** 3 good
**Presentation:** 4 excellent
**Contribution:** 3 good
**Rating:** 6
**Confidence:** 5

**Summary:**

This work introduces Cross-Image Object-Level Bootstrapping (CrIBo), a self-supervised visual representation pre-training approach that enforces object-level representation consistency across different images. Different from existing self-supervised pre-training strategies that only considers image-level representation consistency or cross-view object-level representation consistency, this work also aligns representations of semantically similar objects from different images. By introducing this cross-image object-level self-supervision, CrIBo pre-trains Vision Transformers (ViT) with improved visual representation quality, as demonstrated by the semantic segmentation experiments in various settings, including nearest neighbor retrieval, linear probing, and fine-tuning.

**Strengths:**

- For the first time, this method introduces representation consistency of semantically similar objects from different images into self-supervised pre-training. This would be helpful for performing pre-training on image datasets of complex scenes.

- By leveraging the proposed cycle-consistency condition, trivial matchings between objects can be avoided and the visual encoder is enforced to learn more abstract semantic representations. This design can be helpful for future self-supervised pre-training approaches.

- The writing and figure illustrations are clear. I enjoyed reading this paper.

**Weaknesses:**

- Inadequate experiments: The experiments only consider the semantic segmentation task, which is just one example of object-centric visual recognition tasks. It is strongly suggested to show the generalizability of CrIBO to object detection, instance segmentation, panoptic segmentation, etc.

- Efficiency comparison: Since CrIBO introduces additional operations like clustering and NN retrieval, it is suggested to compare CrIBO and previous methods in terms of the training efficiency. For example, time per epoch may be listed in Table 1.

- Fixed-$K$ clustering: The clustering algorithm (Sec 3.2.1), which comes from a previous work CrOC, is adopted here to identify $K$ objects and $K$ object-level representations in each view. My concern with this clustering algorithm (and other similar methods like K-means as in ODIN [1]) is that it always generates a fixed number of objects per image, regardless of the contents in the image under consideration. One simple image may contain less than $K$ objects; another complex scene may consist of much more than $K$ objects. Thus, it may be complicated to decide an optimal $K$ for a given pre-training dataset. Furthermore, even though an optimal $K$ is found, it can be sub-optimal for some images in the dataset. Would it be possible to improve over such fixed-$K$ clustering algorithms?

- Early-stage training signals: As introduced in Sec 3.2.2, a cycle-consistency criterion is applied to decide whether a pair of object representations is adopted in training. I would assume that at the early training stage, the representations are nearly random since the visual encoder is just initialized. Therefore, there may be only few or even zero object pairs that can pass the test. If I understand Figure 5 correctly, it supports this assumption. Would this phenomenon significantly reduce the available training signals and slow down the early-stage learning?

[1] Olivier J. Hénaff, Skanda Koppula, Evan Shelhamer, Daniel Zoran, Andrew Jaegle, Andrew Zisserman, João Carreira, Relja Arandjelović. Object discovery and representation networks. In ECCV, 2022.

**Questions:**

- Sec 3.1: In the description of object representation, what does “aggregating” exactly mean? Average pooling?

- Sec 3.2.2: When the memory bank is full, are the oldest samples replaced? Or is there a more sophisticated replacement strategy?

- Table 3: In this fine-tuning experiment, CrIBo pre-trained ViT-B underperforms ViT-S. This trend is different from previous experiments. Why?

---

> ### Author Response · Authors · 2023-11-21
>
> ### Thank you for your thorough review and thought provoking questions. Below, we answer all of your concerns / questions.
>
> > Inadequate experiments
>
> We acknowledge the potential benefits of additional downstream tasks to gain deeper insights into the inherent properties of various pretraining schemes, and we recognize the relevance of the listed tasks to our study. However, we would like to address the characterization of our experiments as "inadequate". This term is in our opinion too stringent, given that among the compared methods, only one (MAE) provides metrics for object detection and instance segmentation, and none incorporate panoptic segmentation in their evaluations. This context highlights that our initial focus on semantic segmentation aligns with common practices in the field. Nonetheless, following your suggestion, we have now added an object detection downstream evaluation using YOLO-S (see Tab. 5 and Tab. 6).
>
> > Efficiency comparison
>
> A high-level profiling of CrIBo can be found in Appendix B.7 of the revised manuscript. Our findings indicate that the additional operations specific to object-level nearest neighbors account for less than 15% of the total time. This is a relatively modest increase in computational demand, considering the significant enhancements in performance that CrIBo offers.
>
> > Fixed-K clustering
>
> We agree that a clustering algorithm that adaptively finds the optimal number of clusters for each image is more elegant. Interestingly, the original clustering algorithm from CrOC is adaptive as it relies on the optimal transportation cost to determine the number of centroids. Therefore an "improved"/adaptive version of the clustering already exists.
>
> However, we intentionally decided to deviate from the adaptive scenario for two reasons. Firstly, the adaptive version is more computationally involved as it is iterative and requires running the algorithm with multiple values of K. Secondly, we observed on-par or better results in the overclustering regime with a fixed number of clusters compared to the adaptive version.
>
> Moreover, suggestions from R4 led us to inquire about using supervised annotations instead of our unsupervised clustering. The results from Appendix B.8 show that the overclustering regime is indeed superior rather than the "perfect adaptivity" setting where K is set to the number of objects in the image (provided by the labels). As a final note, it should be observed that even when the initial number of centroids is fixed, the algorithm still bears some adaptiveness due to the pruning of clusters that do not span both views.
>
> > Early-stage training signals
>
> Indeed, Figure 5 supports this assumption. However, the cycle-consistency criterion is only applied to the cross-image object-level loss, as detailed in Equation 8. As such, the training signal from the two other losses is left unchanged. Therefore, in the early-stage of learning, CrIBo benefits from as much training signals as methods that rely exclusively on cross-view objectives (e.g. CrOC). Moreover, the additional training signal provided by the cross-image loss becomes increasingly significant as the training progresses. This phased approach ensures that the self-supervised training signal from the cross-image loss is harnessed only when it is beneficial.
>
> > Object representation aggregation?
>
> In the scope of CrIBo, object representations are computed by average pooling local representations associated with each object. However, this is a design choice, and the aggregation could be done differently. For instance, one could use soft-clustering algorithms, where the aggregation is achieved through a weighted sum of all tokens proportionally to their soft assignment to each object.
>
> > Memory bank update?
>
> Indeed, in our implementation, the memory bank functions as a FIFO queue. When the memory bank reaches its capacity, the oldest samples are simply replaced with newer ones.
>
> > Underperforming ViT-B in the fine-tuning evaluation?
>
> We have identified two primary factors contributing to this outcome.
>
> Firstly, owing to constraints in time and computational resources, our hyperparameter grid search was exclusively conducted for the ViT-S/16 on ImageNet-1k. The best-performing hyperparameters from this search were then applied to both the ViT-S/16 and the ViT-B/16, creating an inherent bias in favor of the former. We have now conducted an independent grid search for the ViT-B/16 and report improved scores, which are detailed in the revised manuscript.
>
> Secondly, we posit that the number of pretraining epochs significantly influences the downstream performance of this task. To support this claim, we report  the performance achieved by a ViT-S/16 pretrained for 300 epochs with CrIBo. These additional results underscore the advantage of extended training schedules for optimizing the performance in this task.

---

> > ### Comment · Reviewer_w5sb · 2023-11-22
> > **Reviewer Response**
> >
> > Thank you very much for the detailed explanations and additional experiments. They have cleared most of my previous concerns. Therefore, I would like to update my rating.
> >
> > Just a few quick follow-up questions/suggestions:
> > - In addition to Table 9 (High-level profiling of CrIBo), it may be helpful to directly compare the wall-clock time per training epoch of CrIBo and each baseline, on the same computation platform.
> > - Why is MAE not compared in Table 6 (Object detection with YOLO-S)?
> > - It seems that the optimal hyper-parameters are diverse when the ViT scale and pre-training dataset change (Appendix A.1). Do you observe that the final results are sensitive to those hyper-parameters? Any general guidelines when deciding the hyper-parameters if future practitioners apply CrIBo in pre-training new models on new datasets?

---

> > > ### Author Response · Authors · 2023-11-22
> > >
> > > Thank you for the prompt reply.
> > >
> > > > Comparison of the wall-clock time per training epoch
> > >
> > > We agree that such a comparison would be useful for the community. However, we won't be able to provide this comparison before the end of the discussion period.
> > > Indeed, this requires installing dependencies and running code from numerous online repositories, which is not trivial considering that our platform is AMD/ROCm-based.
> > > As a side note, our implementation is based upon DINO, which discloses its runtimes, thus CrIBo's runtimes are expected to be 1.15x slower than that.
> > >
> > > > Absence of MAE in Table 6
> > >
> > > Owing to constraints in time and computational resources, we decided to only run YOLO-S for ViT-S/16 backbones pretrained on ImageNet-1k and such a checkpoint is not available for MAE.
> > > In particular, we wouldn't have been able to report scores for the ViT-B/16 pretrained with CrIBo in due time as the grid search only finished recently.
> > >
> > > > Sensitivity of the hyperparameters
> > >
> > > We agree that there is some amount of variation in the optimal hyperparameters from one backbone to the other and similarly for different datasets. However, the results are not drastically different (see initial submission before grid search or Table 4). And this also seems to be a common practice in the ML community e.g. in DINO:
> > >
> > > `norm_last_layer`: ViT-B/16 (True), ViT-S/16 (False)
> > >
> > > `warmup_teacher_temp_epochs`: ViT-B/16 (50), ViT-S/16 (30)
> > >
> > > `clip_grad`: ViT-B/16 (0.3), ViT-S/16 (0)
> > >
> > > `freeze_last_layer`: ViT-B/16 (3), ViT-S/16 (1)
> > >
> > > `lr`: ViT-B/16 (0.00075), ViT-S/16 (0.0005)
> > >
> > > `min_lr`: ViT-B/16 (2e-06), ViT-S/16 (1e-05)

---

> ### Author Response · Authors · 2023-11-23
>
> Since R4 is also keen on seeing a time comparison with different methods, we will provide such a table for a limited number of noteworthy methods e.g. DINO and MAE in the camera-ready version.

---

### Official Review · Reviewer_sFy5 · 2023-10-31

**Soundness:** 3 good
**Presentation:** 3 good
**Contribution:** 3 good
**Rating:** 8
**Confidence:** 4

**Summary:**

This paper works on self-supervised learning from scene-centric images, and locates in between the use of object-level contrastive learning and cross-image nearest neighbours. Compared with prior related work Hummingbird, it proceeds on object regions discovered by the network online instead of pixels, enabling strong representations for both in-context learning evaluation and traditional evaluations (linear probing & fine-tuning). A novel cycle-consistency strategy, which requires NN's another view to be of consistent semantic with another view's NN, creates positive pairs with reasonable variation, and helps improve representation. Experiment results are supportive.

**Strengths:**

*Originality*: The core idea of this paper is novel and original. Though the overall framework still fits in common practices of object-level contrastive learning, the integration of nearest neighbours and the design of cycle consistency are elegant and useful.

*Quality*: The proposed method is well-motivated and extensively evaluated. Its relationship to prior works on different topics is also clearly discussed. I also like the discussions on over-clustering and how it helps scene-centric learning (compositionality), which provided some new insight.

*Clarity*: The delivery is very clear and easy to understand, I did not find issues in understanding.

*Significance*: This work is inspiring and could be helpful for both in-context scene understanding and representation learning.

**Weaknesses:**

- (minor) It would be better if a comparison in GPU memory / time cost in both pre-training and in-context inference stages.
- (minor) The comparison of scene-centric learning is limited to those with ViT-based architectures. If possible, other closely related works on this topic should also be discussed and compared (eg, ORL, SlotCon, COMUS). For instance, I find from SlotCon similar visualizations as fig.3, and also similar model-discovered compositional concepts (eg, parts). Is it possible to apply in-context learning evaluation to these prior works?

References:
- ORL: Unsupervised object-level representation learning from scene images, NeurIPS'21
- SlotCon: Self-supervised visual representation learning with semantic grouping, NeurIPS'22
- COMUS: Unsupervised semantic segmentation with self-supervised object-centric representation, ICLR'23

**Questions:**

Typo: "Finetuning with Segmnenter" -> segmenter (page 14)

---

> ### Author Response · Authors · 2023-11-20
>
> (Edit: Appendix B.6 -> B.7)
>
> ### Thank you for your positive review and valuable suggestions! Below we address your suggestions.
>
> > GPU memory / time cost
>
> A high-level profiling of CrIBo (during training) can be found in Appendix B.7 of the revised manuscript. Notably, less than 15% of the total time is taken by operations specific to object-level nearest neighbors. We do not have such a table for the in-context inference as this is not specific to CrIBo but refer you to Hummingbird for more details on the efficiency of the evaluation.
>
> > ORL, SlotCon, COMUS
>
> ORL, SlotCon, COMUS have been added to the "Related Works" section. Additionally, we applied in-context learning to ORL and SlotCon (COMUS does not seem to have public checkpoints). These results can be found in Table 1. Notably, CriBo outperforms both methods.
>
> > Segmnenter typo
>
> This has been corrected in the revised manuscript.

---

> > ### Comment · Reviewer_sFy5 · 2023-11-21
> >
> > The PDF seems to be not yet updated?

---

> > > ### Author Response · Authors · 2023-11-21
> > >
> > > Indeed, we were waiting on some jobs running to reply to other reviewers and finalize the revised draft. It will be uploaded shortly, sorry for the confusion.

---

> > > > ### Comment · Reviewer_sFy5 · 2023-11-22
> > > >
> > > > Thanks to the authors for their response. I have gone through the updated paper and confirm my positive recommendation. Nice work!

---

### Official Review · Reviewer_FBUq · 2023-10-31

**Soundness:** 4 excellent
**Presentation:** 4 excellent
**Contribution:** 4 excellent
**Rating:** 10
**Confidence:** 4

**Summary:**

The authors propose a shift in paradigm for self-supervised learning (SSL) methods; most SSL methods consider global (image-level) representations and are trained on object-centric datasets but the majority existing real-world images are instead composed of multiple objects (scene-centric). Their proposal is to add to the traditional cross-view consistency with cross-view object-level consistency and cross-image object-level consistency, approaching every image as a collection of objects that may also be present in other images. They show state-of-the-art performance on dense nearest neighbors retrieval and segmentation with a linear head and have SoTA-comparable performance when fine-tuned for segmentation.

**Strengths:**

- The manuscript is easy to follow and compelling to read. Preliminaries are helpful and the organization of the design into three steps is easy to follow.
- The authors present a direct improvement on the method of Hummingbird with a more general framework for scene-centric SSL
- The method is the SoTA on dense nearest neighbors retrieval without any fine-tuning steps
- Empirical evidence shows that each addition was beneficial to the final performance; I have found the B.2 and B.4 experiments to be particularly beneficial to support the implementation of the cycle consistency criterium, one of the "riskier"/less intuitive ideas from the paper.

**Weaknesses:**

- While most of the paper is easy to follow, the section on cycle-consistency matching (3.2.2) is particularly convoluted; on a similar note, Figure 3 is not very intuitive on a first glance and it is hard to determine "who is being compared to whom" to determine consistency.
- Experiments on linear segmentation fail to comment on direct backbone comparisons; this is important since it seems that the model is only better than TimeT (Salehi et al., 2023) -- a ViT-S/16 model when using a larger ViT-B/16 backbone; this is probably due to the larger training set used but still deserves a mention.
- Opinion: I disagree that the fine-tuning regime is suboptimal for comparing SSL models since this is the regime that is going to be used by partitioners when enough training data is available; I also do not think lower/comparable performance in one of the downstream tasks detracts from the manuscript.

Extra notes:
- Table 6 has swapped headers (dataset and epochs)

**Questions:**

1. Given that a lot of the design comes from DINO, did the authors make use of the large-crop/small-crop protocol for teacher and student respectively?
2. On a similar note, DINO benefits considerably from using smaller patch sizes (S/8, B/8, ...); did you experiment with this setup?
3. A global representation is also trained but not used in any of the downstream tasks, what do the authors know of the usefulness of this representation? Is it comparable to object-centric SSL representations?

---

> ### Author Response · Authors · 2023-11-20
>
> ### We appreciate the positive review, constructive feedback and the time taken to review our paper! Below, your questions / concerns are all addressed individually.
>
> > Unclear section 3.2.2
>
> We acknowledge your concerns about the clarity of the cycle-consistency matching section (3.2.2) and Figure 3. In the revised manuscript, we split Figure 3 (in initial draft) into two distinct figures for better comprehension (Figure 3 and Figure 4 in the revised manuscript). The first new figure serves as a visual guide to explain the cycle condition mechanism, while the second is simply a visualization of cross-image object-level matches. Additionally, we have revised section 3.2.2 to improve clarity and readability.
>
> > No direct backbone comparisons
>
> To address the need for direct backbone comparisons, we have made a clear distinction between ViT-B/16 and ViT-S/16 models in all our tables. TimeT is indeed a strong baseline, and its performance relative to different model sizes will be more apparent with this new separation. We also emphasized this point in our revised text to ensure a comprehensive understanding.
>
> Additionally, it's important to note that our method, CrIBo, consistently outperforms concurrent methods across all downstream datasets for a given backbone and pretraining dataset in the context of linear downstream segmentation.
>
> > Opinion: fine-tuning regime is NOT suboptimal
>
> Thank you for sharing your perspective on the suitability of the fine-tuning regime for comparing self-supervised learning (SSL) models. We agree with your viewpoint that fine-tuning is indeed a practical approach, particularly when ample training data is available. However, a caveat here is that when (many) learnable parameters are used for the evaluation, it becomes challenging to discern whether the enhanced downstream performance is primarily attributable to the added capacity of the learnable parameters or to the effectiveness of the pretraining itself. This is why we emphasize evaluating the "intrinsic" quality of the learned representations in scenarios where the downstream evaluation involves few or no learnable parameters. In such cases, we believe the downstream performance better reflects the pretraining's properties.
>
> > Table 6 has swapped headers
>
> This has been corrected in the revised manuscript.
>
> > What about multi-crop?
>
> In this work, we exclusively explored the utilization of two global crops, excluding any local crops. This decision was motivated by a strong emphasis on compatibility with scene-centric datasets. Our observations indicated that, in this context, DINO did not derive benefits and, in some cases, experienced a decline in performance when local crops were introduced.
>
> > What about smaller patch sizes?
>
> This question indeed naturally arises but we initially decided not to run models with smaller patch sizes as the paper was mostly focused on the in-context learning part. In-context learning is however dependent on the patch size as the classification is done at the patch level. We thought about concatenating multiple adjacent patches to compare models of different patch sizes on equal ground. However, it seemed cumbersome and potentially a bit over-engineered. Nonetheless, in response to this request, we investigated the utilization of smaller patch sizes for ViT-S/8 (training on COCO) and tested it on downstream linear segmentation. Notably, we observed enhanced performance and have incorporated these changes into the revised manuscript in Table 2.
>
> > What about the global representation?
>
> In response to this request, we examine the learned image-level representation using the standard k-NN evaluation method, as implemented in DINO. To that end, we extend the ablation on the individual contribution of each loss component with the aforementioned evaluation (see Table 5 of the revised manuscript).
>
> Notably, in the context of a pretraining on COCO, both the cross-view and cross-image object-level losses have a positive impact on the image-level representation. We posit that in the case of scene-centric images, object-level consistency objectives are less susceptible to contextual biases that often plague SSL methods, thereby mitigating the well-known issue of entangling the representations of objects frequently appearing together. In the context of a pretraining on ImageNet-1k, CrIBo (ViT-S/16) obtains 70.9% accuracy. As a point of reference, DINO (with 8 local crops) obtains 74.4%.

---

> > ### Comment · Reviewer_FBUq · 2023-11-22
> >
> > Thank you for the thorough response! I have gone through the updated paper and really liked the improvements! The new Figure 3 (and 4) looks great and the distinction between backbones in the tables is immensely helpful.

---

### Author Response · Authors · 2023-11-21

(Edit: 4 "TBD" entries from running jobs in Table 2 and Table 7 have been added. The revised version is now final.)

We thank all reviewers for their time, valuable comments and the overall appreciation of our work! This past week, we have been busy running the multiple experiment requests and apologize for the rather late reply to your reviews. Nonetheless, the empirical results from these additional experiments provide new insights and improved performance for CrIBo.

There was a recurring concern among reviewers about the underperformance of the ViT-B/16 compared to the ViT-S/16 and other baselines. Owing to constraints in time and computational resources, our hyperparameter grid search was exclusively conducted for the ViT-S/16 on ImageNet-1k. The best-performing hyperparameters from this search were then applied to both the ViT-S/16 and the ViT-B/16, creating an inherent bias in favor of the former.

We have now performed a grid search on the hyperparameters of the ViT-B/16 (K=[4, 8, 12, 16, 32] x $\lambda_{\text{pos}}=[1.0, 2.0, 4.0]$) and report significant performance improvements in the finetuning setting, but also in all other downstream tasks.

---

### Meta-Review · Area_Chair_3QK1 · 2023-12-04

**Metareview:**

All reviewers are positive about the contribution of this work.

**Justification For Why Not Higher Score:**

Several concerns brought up by the reviewers as well as the praise

**Justification For Why Not Lower Score:**

All reviewers lean positive

---

### Decision · Program_Chairs · 2024-01-16

Accept (spotlight)